# Clarity: The Flexibility-Interpretability Trade-Off in Sparsity-aware Concept Bottleneck Models

## Abstract

The widespread adoption of deep learning models in computer vision has intensified concerns about interpretability. Despite strong performance, these models are often treated as black boxes, with limited systematic investigation of their decision-making processes. While many interpretability methods exist, objective evaluation of learned representations remains limited, particularly for approaches that rely on sparsity to "induce" interpretability. In this work, we investigate how modeling choices in Concept Bottleneck Models (CBMs) affect the semantic alignment of concept representations. We introduce *Clarity*, a post-hoc diagnostic measure that captures the interplay between downstream performance and the sparsity and precision of concept activations. Using an interpretability assessment framework grounded in datasets with ground-truth concept annotations, we evaluate both VLM- and attribute predictor-based CBMs across three amortized sparsity-inducing strategies ($\ell_1$, $\ell_0$, and Bernoulli-based), alongside several widely used sparsity-aware CBM methods from the literature. Our experiments reveal a critical *flexibility-interpretability trade-off*: a model's capacity to optimize task performance by deviating from semantic alignment. We demonstrate that under this trade-off, different methods exhibit markedly different behaviors even at comparable performance levels. Finally, we validate our framework through a principled human study, which confirms that *Clarity* aligns significantly more closely with human trust than standard evaluation metrics.

## 1 Introduction

The recent advent of large, pre-trained models in computer vision has driven an unprecedented adoption of modern deep architectures in a variety of applications and domains. Unfortunately, these models are frequently treated like black-boxes; they map inputs to outputs with little to no systematic analysis of their decision making process. This lack of transparency raises serious questions about their confident deployment in critical settings, further amplifying the need for transparent and interpretable architectures.

Concept Bottleneck Models (CBMs) (Koh et al., 2020), and concept-based models in general, aim at tackling this problem; these constitute ante-hoc methods, where the aim is to create inherently interpretable models by constructing a *bottleneck*, i.e., a human interpretable layer that is used to generate the final prediction towards the downstream task. CBMs were initially considered using hand-annotated concepts, which are labor intensive and costly or even unfeasible to produce; at the same time, their dependence on a restricted set of concepts led to significant performance degradation. To bypass this limitation, recent works leveraged vision-language models (VLMs), enabling the use of unrestricted concept sets either hand-crafted, automatically extracted or even machine generated via LLMs (Yang et al., 2023; Oikarinen et al., 2023). While this shift paved the way for unrestricted concept sets that are shown to be quite effective for task performance (Oikarinen et al., 2023), it came at the expense of interpretability. Indeed, VLM-based approaches typically consider thousands of concepts, rendering the investigation of the bottleneck a challenging and unintuitive task, while other issues were further intensified, such as information leakage (Margeloiu et al., 2021; Schoen et al., 2025).

Sparsity-aware methods aim at solving this issue (Yang et al., 2023; Oikarinen et al., 2023; Panousis et al., 2023). However, these often fail to properly assess the actual impact on model interpretability, instead

emphasizing metrics such as classification performance and overall sparsity, typically complemented by qualitative visualizations of the learned representations. Even when alignment with ground-truth information is explicitly evaluated, standard multi-label prediction metrics, typically binary accuracy, are often used despite their limitations in sparse settings (Panousis et al., 2024) and without considering their suitability as a proxy to interpretability. In many settings, ground-truth attributes are highly sparse; consequently, methods that predict a large number of inactive concepts can achieve deceptively high binary accuracy. Instead, it is crucial to assess which concepts are correctly inferred as active relative to the total number of active concepts (i.e., the precision of the concepts used towards the downstream task) while simultaneously accounting for the imposed sparsity ratio and the performance of the model toward the downstream task. Thus, in this work, we argue for a return to the initial drive behind concept-based approaches by explicitly analyzing the effect of modeling choices on interpretability and performance. In particular, we are interested in the notion of *flexibility*, defined as the model's capacity to optimize predictive performance while, potentially, deviating from strict semantic alignment. In practice, flexibility manifests itself when models exploit discriminative but semantically incorrect concepts to improve accuracy, effectively decoupling predictive performance from the quality of the underlying concept representation. Understanding and controlling this behavior is central to characterizing the interpretability-performance trade-off in modern CBMs. This motivates the need for a metric that explicitly penalizes such flexible but semantically misaligned solutions. To this end, we propose a new metric to measure the *Clarity* of a concept-based representation; this captures the interplay between three desiderata in CBMs: *high sparsity*, to enable per-example inspection of the used concepts; *high concept prediction precision*, to ensure that the selected concepts are semantically correct while allowing for the selection of informative but non-comprehensive concept-based descriptions; and *strong downstream classification accuracy*, so that interpretability does not come at the expense of the downstream task performance. We consider two distinct CBM formulations: one based on explicit attribute prediction and the other grounded in VLMs, and systematically evaluate the impact of sparsity across three different sparsity-aware methods, while also examining other widely used CBM formulation, such as Post-Hoc (Yuksekgonul et al., 2022) and Label-Free CBMs(Oikarinen et al., 2023). Our results show that interpretability in CBMs is not guaranteed by sparsity or accuracy alone, but requires explicitly accounting for the semantic correctness of the underlying concepts. Our contributions can be summarized as follows:

- We identify and formalize the *flexibility-interpretability trade-off* in concept bottleneck models, showing how modern architectures can achieve high predictive performance by deviating from semantically correct concept representations.

- We propose *Clarity*, a unified metric that captures the interplay between sparsity, concept precision, and downstream accuracy, providing a post-hoc diagnostic measure of the flexibility-interpretability trade-off in concept-based models.

- We demonstrate, through a human evaluation study, that Clarity strongly correlates with user trust, outperforming individual components and standard aggregation schemes.

- We conduct a comprehensive empirical study across diverse CBM architectures, including post-hoc, label-free, and sparsity-aware models, showing that in the considered settings, high accuracy is often achieved at the expense of concept precision.

## 2 Related Work

**Concept Bottleneck Models.** Given a classification problem characterized by a dataset $\mathcal{D} = \{\boldsymbol{X}_n, \hat{\boldsymbol{y}}_n\}_{n=1}^N$ comprising $N$ image/label pairs, where $\boldsymbol{X}_n \in \mathbb{R}^{I_H \times I_W \times c}$ and $\hat{\boldsymbol{y}}_n \in \{0, 1\}^C$, CBMs (Mahajan et al., 2011; Koh et al., 2020) aim at improving the interpretability of vision models by first predicting a vector $\mathbf{a}_n = f_\theta(\boldsymbol{X}_n) \in [0, 1]^M$ of concept scores; these correspond to a *concept set* $A = \{a_1, \ldots, a_M\}$, where every element is associated with a human-understandable concept. The score is related to the likelihood of the presence of such concept in the image, followed by a linear layer that maps the concept scores to the downstream task, $\mathbf{y}_n = \mathbf{W}_c^\top \mathbf{a}_n$. Training the concept predictor requires a dataset of image-concept labels, which has motivated the emergence of label-free approaches (Oikarinen et al., 2023; Yang et al., 2023) that leverage pretrained VLMs, e.g., CLIP (Radford et al., 2021), to estimate the presence of concepts in an image.

**Leakage in Concept Bottleneck Models.** Information leakage, a phenomenon in which the model hijacks the concept bottleneck to directly encode information for the downstream task, is known to pose a big problem to the interpretability of CBMs (Margeloiu et al., 2021; Schoen et al., 2025), and can stem either from wrong concept predictions in the bottleneck or from nonsensical relations between the concepts and the downstream task. Leakage has been shown to be difficult to measure (Zarlenga et al., 2023), with some authors proposing to use occlusion of ground truth object parts to detect and correct it (Huang et al., 2024). Some methods to prevent leakage involve adding a non-interpretable side channel to the model (Havasi et al., 2022), releasing pressure from the concept predictor but turning the model into a partial black-box, or carefully choosing the concept set (Ruiz Luyten & van der Schaar, 2024), which is not always an option in real-world applications. In this work, we argue that one common manifestation of leakage arises when a model maintains high accuracy and sparsity while relying on semantically incorrect concepts. In such cases, predictive performance is effectively decoupled from the intended semantic bottleneck, leading to behavior that is functionally equivalent to leakage at the representation level. This observation suggests that the degree of semantic misalignment in the active concepts provides a measurable proxy for this failure mode, which we formalize later through the Clarity metric.

**Attribute-based Zero-shot Learning.** CBMs share a direct lineage with early attribute-based Zero-Shot Learning (ZSL) methods, which similarly decompose prediction into an image-to-attribute stage followed by an attribute-to-label mapping (Lampert et al., 2009; Xian et al., 2019). These approaches, such as Direct Attribute Prediction (DAP) (Lampert et al., 2009), introduced intermediate semantic attribute layers to decouple image features from final labels, building cross-category robustness. However, as ZSL moved toward joint embedding methods like Attribute Label Embedding (ALE) (Akata et al., 2013) to optimize global classification performance, a persistent trade-off emerged: downstream task gains frequently came at the direct expense of individual attribute accuracy, thus deviating from true semantic grounding. For instance, classifiers routinely achieve high accuracy on localized or non-visual traits not by detecting the visual property, but by leveraging correlated contextual cues (Lampert et al., 2014). Consequently, visual attributes have long been recognized as highly unreliable signals in purely discriminative settings, prompting classical ZSL architectures to explicitly model prediction uncertainty (Jayaraman & Grauman, 2014) or enforce strict "semantic-preserving" constraints (Jiang et al., 2017). In this sense, the semantic alignment problems studied in modern CBMs are closely related to longstanding issues in attribute-based recognition. We demonstrate that this historical lineage of shortcut exploitation and semantic drift has returned within modern VLM-based CBM implementations. Despite utilizing massive pre-trained parameter spaces, contemporary models systematically repurpose labeled concepts into ungrounded, discriminative features to maximize downstream performance (Debole et al., 2026). We show that models can maintain high predictive performance despite substantial concept-level misalignment, and we introduce Clarity as a diagnostic metric for measuring this phenomenon through the joint evaluation of accuracy, sparsity, and concept precision.

**Sparsity in Concept Bottleneck Models.** On top of raising concerns about the difficulty of learning a concept predictor, Ramaswamy et al. (2023) point at another overlooked problem: large concept sets can render CBMs actually unintelligible to humans, with users preferring a handful of concepts per explanation. This has driven the development of sparsity-aware CBMs, where only a few concepts are selected to contribute to the downstream task, while the rest are effectively zeroed out. These methods focus on sparsifying either the concept bottleneck scores (Panousis et al., 2023; 2024) or the concept-class matrix that links concept scores to the downstream task (Yuksekgonul et al., 2022; Yang et al., 2023; Oikarinen et al., 2023; Schrodi et al., 2025), while some aim at inducing both types of sparsity simultaneously (Marcos et al., 2020).

## 3 Proposed Framework

In this work, we focus on Concept Bottleneck Models and specifically concept-based classification. Let us denote by $\mathcal{D} = \{\mathbf{X}_i, y_i\}_{i=1}^{N}$, a dataset comprising $N$ examples, each belonging to a class $c \in C$, such that $y_i \in \{0, 1\}^C$. We additionally consider a *concept set* $A = \{a_1, \ldots, a_M\}$, which typically comprises human-interpretable notions, usually expressed by text or attribute presence. Within this frame of reference, we first aim to train a predictor that outputs the concept scores for each image, which is then used to classify each example; thus, we create an interpretable bottleneck to drive the downstream task. Assuming a transformation $f_{\text{pred}}(\boldsymbol{X}_i) \in (0, 1)^M$ that predicts, for each image, the concept scores, and a classification

weight matrix $\boldsymbol{W}_c \in \mathbb{R}^{M \times C}$, the optimization process reads:

$$\min_{W_c} \mathcal{L}_c = \sum_{i=1}^{N} \text{CE}(\boldsymbol{W}_c^\top f_{\text{pred}}(\boldsymbol{X}_i), y_i) \tag{1}$$

The predictor is commonly a neural network that is pre-trained by minimizing the binary cross-entropy between its sigmoided predictions and the ground-truth concept labels; during concept-based classification, it remains frozen.

However, this formulation of CBMs is considered very restrictive. Indeed, annotating concepts for a particular dataset and concept set is a very strenuous process, while the presence or not of concepts is often open to human interpretation. To this end, several works aim to bypass the need for ground-truth information by employing VLMs (Yang et al., 2023; Oikarinen et al., 2023; Panousis et al., 2023). In this setting, we do not need to train a predictor; instead, by exploiting the zero-shot capabilities of VLMs we can obtain the similarity between any image in the dataset and any potential concept set. Denoting by $E_I(\boldsymbol{X}_i) \in \mathbb{R}^K$, $E_T(\boldsymbol{A}) \in \mathbb{R}^{M \times K}$, the $K$-dimensional embeddings of the image and of the concept set, respectively, obtained through the encoders of a considered VLM, the concept scores can be predicted as:

$$f_{\text{VLM}}(\boldsymbol{X}_i) = E_I(\boldsymbol{X}_i) E_T(\boldsymbol{A})^\top \in \mathbb{R}^M \tag{2}$$

with the classification loss being similar to Eq. 1, where instead of $f_{\text{pred}}$, we use $f_{\text{VLM}}$.

### 3.1 Sparsity-aware Model Formulation

Although VLMs have considerably increased the flexibility of CBMs, at the same time, they have harmed the interpretability efforts by allowing the usage of massive concept sets while basing the inference process on the implicit concept similarity provided by them. To this end, and following relevant VLM-based CBM literature (Yang et al., 2023; Oikarinen et al., 2023; Panousis et al., 2023; 2024), we consider a sparsification approach, aiming to drastically reduce the *per-example* number of active concepts to facilitate interpretability of the individual emerging representations.

To this effect, we consider auxiliary binary latent variables $\boldsymbol{z}_i \in \{0,1\}^M, \forall i$; these denote the presence or absence of the concepts $\{a_m\}_{m=1}^M \in \boldsymbol{A}$ for each example, by masking the corresponding entries during classification, i.e.,

$$\hat{\boldsymbol{y}}_i = (\boldsymbol{z}_i \cdot \boldsymbol{W}_c)^\top f_{\text{pred/VLM}}(\boldsymbol{X}_i) \tag{3}$$

We can rely on different estimation approaches to infer the auxiliary latent variables $\boldsymbol{Z}$. Ideally, these latent variables should be highly sparse, yielding a representation that is both compact and potentially interpretable; thus, we want to induce sparsity that not only facilitates the downstream task but also enables a per-example concept investigation and analysis of the emergent behavior of concept selection. To infer these latent variables efficiently, we build on three sparsification paradigms: deterministic $\ell_1$ regularization, continuous $\ell_0$ relaxations via the Hard Concrete distribution (Louizos et al., 2018), and stochastic Bernoulli formulations via mean-field variational inference (Panousis et al., 2023). Rather than optimizing static parameters, we introduce an amortized version for each formulation by utilizing a single amortization matrix $\boldsymbol{W}_s \in \mathbb{R}^{M \times K}$ driven by the VLM's image embeddings. This yields the *unnormalized instance concept scores* $\hat{\boldsymbol{\Phi}} = E_I(\boldsymbol{X}) \boldsymbol{W}_s^\top \in \mathbb{R}^{N \times M}$, which are mapped to the gating variables $\boldsymbol{z}_i$ according to the chosen sparsity mechanism considered.

For the $\ell_1$-based method, we transform these logits via a sigmoid transformation $\sigma(\cdot) = \text{Sigmoid}(\cdot)$, s.t.:

$$\boldsymbol{Z}_{\ell_1} = \sigma(\hat{\boldsymbol{\Phi}}) \tag{4}$$

For the $\ell_0$, we can rewrite the sampling procedure as:

$$u = \mathcal{U}(0,1), \ s = \sigma((\log u - \log(1-u) + \hat{\boldsymbol{\Phi}})/\beta)$$
$$\bar{s} = s(\zeta - \gamma) + \gamma \tag{5}$$
$$\boldsymbol{Z}_{\ell_0} = \min(1, \max(0, \bar{s}))$$

And finally, for the Bernoulli-based procedure:

$$\boldsymbol{Z}_{\text{Bernoulli}} = \text{Bernoulli}\left(\boldsymbol{Z}|\sigma(\hat{\boldsymbol{\Phi}})\right) \tag{6}$$

The corresponding penalty terms and optimization details for each distributional relaxation are provided in Appendix A for completeness. By optimizing this formulation end-to-end, the model learns to identify a sparse, highly predictive subset of concepts for each instance; we now turn to the training procedure and describe how the latent variables and model parameters are jointly learned.

**Training.** We consider two alternative backbones for computing concept scores. One is based on a VLM, where concept scores are directly obtained using Eq. 2, and the other relies on an *attribute predictor* as described in Eq. 1. Specifically, we train a predictor model that takes as input an image's embedding stemming from the image encoder of a VLM and outputs the concept scores; it comprises a single linear layer $\boldsymbol{W}_{\text{pred}} \in \mathbb{R}^{K \times M}$ followed by a sigmoid non-linearity, while optimizing the binary cross entropy between the predictions and the ground-truth attribute presence information.

Given a trained predictor or a VLM that outputs the concept scores for each example, all considered sparsity-aware concept-based models comprise two learnable parameters: (i) the amortization matrix $\boldsymbol{W}_s \in \mathbb{R}^{M \times K}$ and (ii) the classification matrix $\boldsymbol{W}_c \in \mathbb{R}^{M \times C}$. For each method, the final loss involves two terms, the classification loss and the regularization/penalty term stemming from the corresponding sparsification process; these details can be found in the Appendix.

To infer the latent variables $\boldsymbol{Z}$ for each method, we only use the classification signal stemming from their corresponding loss optimizing both $\boldsymbol{W}_s$ and $\boldsymbol{W}_c$ in an end-to-end fashion; then, we freeze the amortization matrix $\boldsymbol{W}_s$ and re-train the classification matrix $\boldsymbol{W}_c$ with different *thresholds* to properly assess the potency of the sparse codes and obtain the final classification performance as detailed next.

**Inference.** Having trained $\boldsymbol{W}_s$, we can compute, for each method, the active concepts per example. Specifically, we introduce a threshold $\tau$ that can be used to determine if a concept is active or not for an example $i$, via its inferred auxiliary variable $\boldsymbol{z}_i$. For $\ell_1$, we can directly threshold the indicators $\boldsymbol{Z}_{\ell_1}$ computed via Eq. 4, s.t.:

$$\tilde{z}_{\ell_1}^m = \begin{cases} 1, & \text{if } [\sigma(E_I(\boldsymbol{X}_{\text{test}})\boldsymbol{W}_s^{\top})]_m > \tau \\ 0, & \text{otherwise} \end{cases} \tag{7}$$

For $\ell_0$, the logit transformation during inference reads:

$$\tilde{\boldsymbol{z}}_{\ell_0} = \min(1, \max(0, \sigma(E_I(\boldsymbol{X}_{\text{test}})\boldsymbol{W}_s^{\top}))(\zeta - \gamma) + \gamma) \tag{8}$$

Thus, we can use $\tau$ as before. For the Bernoulli-based formulation, we can directly threshold the inferred amortized probabilities of the Bernoulli distribution, leading to a similar thresholding rule based on the sigmoided inner product between the amortization matrix and the test input.

## 3.2 The *Clarity* metric

We argue that, for a concept-based model to be truly interpretable, three criteria must be simultaneously satisfied: *high sparsity*, to enable per-example inspection of the active concepts; *high precision*, to ensure that the selected concepts are semantically correct; and *strong classification accuracy*, so that interpretability does not come at the expense of performance in the context of the downstream task. To capture the interplay and trade-offs among these competing objectives, we introduce the notion of *Clarity*, a new diagnostic metric aiming to quantify the flexibility-interpretability trade-off in concept-based models.

Let $X_i$ be an input instance, $M$ be the total number of concepts in the bottleneck, and $z_i^m \in [0, 1]$ be the continuous activation (or probability) predicted by the model for the $m$-th concept. For a given threshold $\tau$, we define the *binarized indicator* of concept activation as:

$$\tilde{z}_i^m(\tau) = \mathbb{I}(z_i^m \geq \tau) \tag{9}$$

where $\mathbb{I}(\cdot)$ is the indicator function. Let $g_i \in \{0,1\}^M$ represent the ground-truth concept annotation vector for instance $\boldsymbol{X}_i$, where $g_i^m = 1$ if the concept is present and 0 otherwise. Then, we can define the three underlying components as:

1. Sparsity: Measured as the *fraction of inactive concepts* per example. For a single instance $\boldsymbol{X}_i$ at threshold $\tau$:

$$Sparsity(\boldsymbol{X}_i, \tau) = 1 - \frac{1}{M}\sum_{m=1}^{M}\tilde{z}_i^m(\tau) \in [0,1] \tag{10}$$

   where the dataset-level sparsity $Sparsity(\tau)$ is the average across all $N$ instances.

2. Precision: Evaluates the semantic accuracy of the *active* concepts against the ground truth. For a single instance $\boldsymbol{X}_i$ at threshold $\tau$:

$$Precision(\boldsymbol{X}_i, \tau) = \frac{\sum_{m=1}^{M}\tilde{z}_i^m(\tau)\cdot g_i^m}{\sum_{m=1}^{M}\tilde{z}_i^m(\tau)} \in [0,1] \tag{11}$$

   The dataset-level precision $Precision(\tau)$ is the average precision across all instances. Within this context, if a model predicts zero active concepts for a given instance $\boldsymbol{X}_i$, we assign a default value of 0, thus systematically pulling down the global dataset-level precision score. Omitting these instances instead of zeroing them out would introduce a severe selection bias, resulting in an artificially inflated and unrepresentative summary of the model's explanatory coverage across the dataset.

3. Accuracy: Represents the standard downstream task classification performance, e.g., top-1 accuracy, achieved by the predictor layer using only information pertaining to the active concepts as described in Eq. 3. Thus, for an input instance $\boldsymbol{X}_i$, let $y_i$ be the ground-truth class label, and let $\hat{y}_i$ be the model's predicted class label score vector. The final predicted class is determined via the argmax over the predictor logits:

$$\hat{y}_i(\tau) = \arg\max_{c}\left[\left(\tilde{\boldsymbol{z}}_i(\tau)\cdot\boldsymbol{W}_c\right)^{\top}f_{\text{pred/VLM}}(\boldsymbol{X}_i)\right] \tag{12}$$

   the dataset-level accuracy $\mathcal{A}(\tau)$ is computed as the fraction of correctly classified instances:

$$Accuracy(\tau) = \frac{1}{N}\sum_{i=1}^{N}\mathbb{I}\left(\hat{y}_i(\tau) = y_i\right) \tag{13}$$

With these definitions at hand, we define *Clarity* as the harmonic mean of these three components (omitting the explicit dependence on $\tau$ for brevity):

$$\text{Clarity} = \frac{3\cdot Sparsity\cdot Precision\cdot Accuracy}{(Precision\cdot Accuracy) + (Sparsity\cdot Accuracy) + (Sparsity\cdot Precision)} \tag{14}$$

**Properties.** Clarity satisfies several desirable properties. (i) *Monotonicity:* it is monotonically increasing in each of *Sparsity*, *Precision*, and *Accuracy*. (ii) *Bottleneck sensitivity:* as a harmonic mean, it is dominated by the smallest component, penalizing imbalanced trade-offs. (iii) *Degeneracy:* if any component approaches zero, Clarity also approaches zero, reflecting a failure in at least one critical dimension of interpretability or performance.

A standard alternative to a unified scalar metric is to report components independently or analyze them through a multi-objective Pareto frontier. While Pareto analysis is useful for understanding the trade-off landscape within a model family, multi-objective methods generally assume a *compensatory* relationship between objectives, where strong performance in one area can offset weaker performance in another. For concept bottleneck models, however, representation integrity does not follow this logic. Downstream accuracy, sparsity, and semantic precision are all essential properties, and shortcomings in any one of them cannot be fully compensated for by improvements in the others. For example, a model may achieve near-perfect

classification accuracy and exceptionally high sparsity, yet exhibit zero concept precision; the same holds for all potential individual metric combinations. In such a case, the model has effectively lost the interpretability guarantees that motivate the concept bottleneck framework. Evaluating metrics in isolation can therefore obscure localized catastrophic failures. Consequently, a model may appear to occupy an advantageous position on the Pareto frontier despite exhibiting severe, or even complete, breakdowns in semantic alignment for particular inputs. By collapsing these elements into a single scalar via the harmonic mean, Clarity enforces a strict "weakest-link" penalty that cannot be bypassed by maximizing a single dimension. This property is particularly important for characterizing the *flexibility-interpretability trade-off*. While a model may leverage its architectural *flexibility* to sustain high accuracy via semantically incorrect but discriminative concepts, such behavior inevitably triggers a significant drop in *Precision*. By employing a harmonic mean, *Clarity* ensures that this opportunistic flexibility is penalized rather than rewarded, thereby distinguishing semantically grounded models from those that rely on spurious concept-task alignments. In this context, low concept precision reflects semantic misalignment between the selected concepts and the ground-truth attributes. Following the discussion in Section 2, we interpret such misalignment as a measurable proxy for leakage-like behavior at the bottleneck level, where the model relies on spurious or semantically incorrect concepts to sustain predictive performance. In the following, we use this metric to investigate how different sparsity-aware formulations navigate this trade-off within the CBM framework.

## 4    Experimental Evaluation and Discussion

**Experimental Setup.**    We focus on the CUB (Wah et al., 2011) and SUN (Xiao et al., 2010) datasets as they provide the "gold-standard" per-instance ground-truth concept annotations required to calculate objective Precision; a metric fundamentally unattainable on larger, automatically-annotated benchmarks that lack human-verified labels. These datasets span diverse characteristics, including number of examples, classes and attributes; CUB comprises 11780 examples divided into 200 classes with per-example ground-truth attributes from a pool of 312, while SUN comprises 14340 examples, spanned across 700 classes with a pool of 102 attributes. CUB attributes are highly visual attributes, e.g., *wing color black, underparts color orange*; the same holds for SUN, although it also includes a few non-visual cues, e.g., *scary, research, stressful*. For each dataset, ground truth information is provided via per-example attribute presence information. When using a VLM to compute the relation between the images and the attributes, we consider as concept set the ground truth attributes provided in each dataset; thus, in this case we consider no explicit ground-truth information and rely on the zero-shot capabilities of VLMs as commonly done in the related literature. We split the CUB dataset according to the typical split, i.e., 5990 examples for training, 5790 for testing, while for SUN we choose a $80\% - 20\%$ split, the same for all methods.

**Training Details.**    To train the *attribute predictor* and drive the computation of the auxiliary binary indicators $\boldsymbol{Z}$ for each of the considered methods, we exploit image embeddings stemming from the image encoder of a VLM, and specifically CLIP. We consider two commonly used CLIP variants, i.e., ViT-B/16 and ViT-L/14; this decision also facilitates the investigation of how different embeddings affect the inferred representations. To avoid re-computing the embeddings at every iteration, we pre-compute them and store them in an efficient format for repeated use. For the *predictor-based* models, we train the attribute predictor for a fixed number of epochs for all combinations of datasets and embedding backbones. For the *VLM-based* models, we compute similarities with text embeddings corresponding to the considered concept set. In all cases, the training of the binary indicators $\boldsymbol{Z}$ is performed by minimizing the corresponding sparsity-aware loss for each method over 1500 epochs across all experimental settings. After learning the amortization matrix for each method, we re-train the classification layer, i.e., $\boldsymbol{W}_c$, using different thresholds and for 200 epochs each, to evaluate how effectively the sparse codes support classification, while ensuring a fair basis for computing the interpretability metrics.

**Threshold Selection Strategy:**    Because the three components of the Clarity metric, namely dataset-level sparsity, precision, and classification accuracy, exhibit non-linear trade-offs as a function of the concept activation threshold $\tau$, a joint optimization is required. For instance, setting an excessively low $\tau$ yields dense bottlenecks that compromise semantic precision, whereas an overly conservative high $\tau$ risks empty representations that catastrophically penalize downstream accuracy. To resolve this equilibrium, we evaluate

the metric across a uniform grid search over the range $\tau \in [0, 1]$ with a step resolution of $\Delta\tau = 0.05$. The optimal operating threshold $\tau^*$ is selected by maximizing the joint metric on validation data:

$$\tau^* = \arg\max_\tau \text{Clarity}(\tau) \tag{15}$$

**Baseline Results.** We begin our experimental result analysis with the assessment of the concept prediction backbones. In Table 1, the attribute prediction performance for the two types of concept score prediction models, Predictor-based and VLM-based, across the two benchmark datasets and using the two backbone architectures for obtaining the embeddings, are reported. For the evaluation metrics in this context, we consider the mean Average Precision (mAP), which captures the overall ranking quality of the predicted attributes, and the Area Under the ROC Curve (AUC).

Table 1: Attribute prediction performance with respect to ground-truth attribute information. We use only example-wise attribute information for training the attribute predictor, while for CLIP we consider the similarity between the image and text embeddings.

| | | ViT-B/16 | | ViT-L/14 | |
|---|---|---|---|---|---|
| **Model** | **Dataset** | mAP ↑ | AUC ↑ | mAP ↑ | AUC ↑ |
| Predictor | CUB | 61.05 | 0.882 | 61.19 | 0.859 |
| | SUN | 73.60 | 0.953 | 74.30 | 0.954 |
| VLM | CUB | 16.32 | 0.540 | 14.91 | 0.565 |
| | SUN | 31.82 | 0.748 | 29.96 | 0.734 |

Therein, we observe that predictor-based models consistently outperform their VLM-based counterparts across both datasets and backbone architectures. This is expected, as models trained with explicit attribute supervision produce more accurate attribute predictions compared to the zero-shot capabilities of VLMs in these settings. Both methods perform better on the SUN dataset than on CUB, reflecting on the nature of the two datasets: SUN comprises 102 scene-level attributes, while the 312 CUB attributes are more fine-grained and harder to discriminate. Notably, for CUB, the VLM-based approach achieves near-random performance, highlighting a potential limitation of its zero-shot properties in specialized, fine-grained domains. Finally, the results suggest that backbone size has limited impact on the final performance for both methods, indicating that increasing model capacity does not yield substantial improvements.

**Concept Selection with Sparsity-aware Methods.** We begin our analysis by examining classification performance as a function of sparsity for both predictor- and VLM-based approaches, depicted in Fig. 1. Therein, the `Attribute Predictor` models are baseline models, where we train a linear layer on top of the attribute predictor yielding an original dense CBM model, similar to (Koh et al., 2020); to evaluate this setting at different sparsity levels, we threshold the concept scores and re-train the classifier. We follow a similar procedure for the `VLM` baselines.

We observe that predictor- and VLM-based methods achieve comparable classification accuracy, despite the previously noted deficiencies in VLM-based attribute prediction. This discrepancy underscores the inherent *flexibility* of the concept bottleneck: models can maintain high task performance by exploiting discriminative but semantically unaligned signals, even when the underlying concept precision is poor. This behavior exemplifies the decoupling described in Section 3, and is consistent with the leakage-like failure mode discussed earlier, where the model relies on semantically misaligned concepts and thereby undermines the intended role of the bottleneck.

In this setting, using a more expressive embedding backbone leads to a noticeable improvement in task performance, likely due to its increased representational capacity and ability to encode richer semantic information. Moreover, all sparsity-inducing methods achieve comparable classification performance at comparable sparsity levels, suggesting that the choice of sparsification approach has limited impact on task accuracy once sparsity is controlled. Importantly, all sparsity-aware methods substantially outperform a naive thresholding of the raw concept scores-for example, `VLM w/ ViT-B/16`. In the same figure, we plot the precision as a function of sparsity. A consistent pattern emerges across all methods: increasing sparsity generally leads to

higher precision, indicating that the concept selection mechanisms assign higher scores to correctly predicted concepts. The Bernoulli-based formulation exhibits a slight advantage, while the choice of backbone has only a modest effect on precision, suggesting that backbone capacity is not a major determinant in this setting.

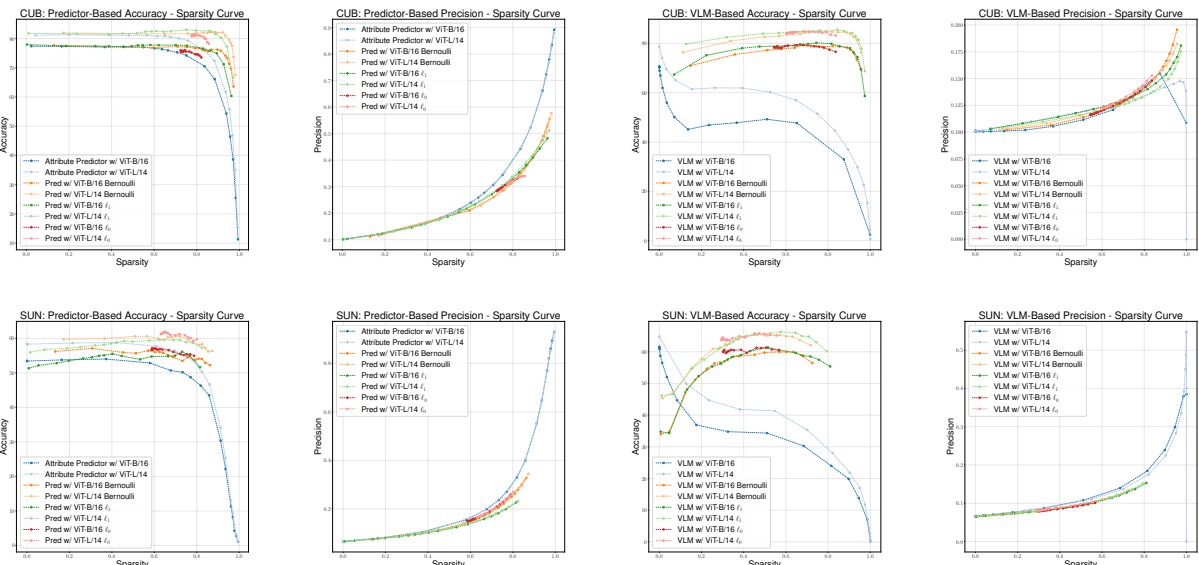

Figure 1: Accuracy-Sparsity and Precision-Sparsity Curves for the CUB (top) and SUN (bottom) datasets. From left to right, we report results for predictor-based methods (first two plots) and VLM-based methods (last two plots). For both methods, all sparsity-inducing approaches maintain strong classification performance even at high sparsity levels. For the VLM-based setting, we observe a significant increase in classification accuracy when employing sparsity-aware methods. This suggests that, despite the limited zero-shot attribute prediction capabilities of CLIP in this scenario, appropriate concept selection can recover substantial discriminative power for the downstream task. At the same time, increased sparsity consistently leads to higher precision in the resulting representations.

However, from these plots, it is not straightforward to determine which model is "best" in terms of interpretability. This is where the notion of Clarity becomes crucial. In Fig. 2, we plot Clarity as a function of task performance, revealing a striking pattern: while all methods can achieve similar task accuracy, they do so at widely different levels of Clarity. This indicates that achieving strong downstream performance does not guarantee interpretable representations, and highlights the importance of explicitly evaluating interpretability metrics such as sparsity and precision alongside task accuracy when comparing sparsity-aware methods. This empirical pattern further illustrates the limitations of traditional Pareto-front analysis. Methods that appear competitive when evaluated, e.g., only through accuracy-sparsity trade-offs can exhibit substantial degradation in concept precision, a failure mode that remains invisible on the Pareto frontier. By incorporating semantic precision directly into a single non-compensatory metric, Clarity exposes these failures, providing a more accurate characterization of representation quality. Ideally, a method should lie in the upper-right region of such a graph, corresponding to high sparsity, high precision, and strong classification performance. In this instance, the Bernoulli-based method performs favorably across most settings, with the exception of the VLM-based formulation on the SUN dataset, where it is outperformed by the $\ell_1$ regularization. Importantly, we do not advocate for a single best-performing method. Different sparsity-inducing approaches may be more optimally constructed or tuned to achieve higher Clarity under specific modeling choices. While we explored a range of configurations for all methods, the reported results are not exhaustive and should be viewed as indicative rather than definitive.

To better quantify differences in model interpretability, Table 2 reports the results for all methods based on the maximum achieved Clarity. We report classification accuracy, sparsity (measured as percentage of inactive concepts per example), and precision. These results highlight the importance of evaluating inter-

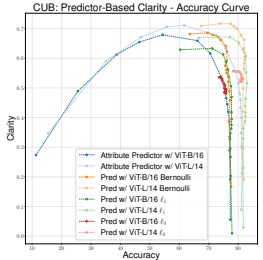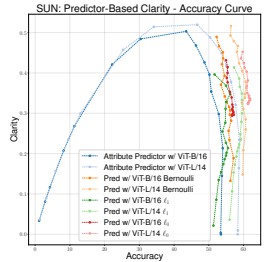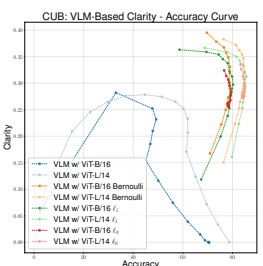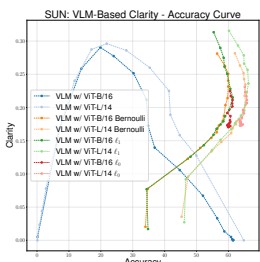

Figure 2: Clarity-Accuracy Curves for both datasets. From left to right, we report the results for predictor-based methods (first two plots) and VLM-based methods (last two plots). While all methods can achieve similar classification accuracy, they do so at widely different Clarity levels, highlighting that strong task performance does not guarantee interpretable representations.

pretability with metrics beyond just sparsity and task performance. In many cases, the model selected for maximum Clarity is not the one with the highest classification accuracy. For instance, the predictor-based Bernoulli configuration achieves the highest Clarity on CUB; although it does not attain the best accuracy or sparsity overall, it achieves the highest precision while balancing all relevant metrics. Conversely, for the VLM-based $\ell_1$, the model selected for maximum Clarity may be sub-optimal in terms of classification accuracy, yet it exhibits superior sparsity and precision, yielding higher overall Clarity than the $\ell_0$ counterpart, which, despite strong task performance, activates more concepts per example and achieves lower concept precision. In the same table, we also report the computed binary accuracy, a metric often used to evaluate interpretability in concept-based models. We observe that binary accuracy can be misleading: in many cases, models achieve similar binary accuracy despite exhibiting substantial differences in key interpretability components, such as precision and sparsity. This is particularly evident for the VLM-based methods, where high binary accuracy does not necessarily correspond to a model with meaningful or sparse concept activations.

Table 2: Results of sparsity-inducing methods on CUB and SUN datasets. We report models with the highest Clarity.

| Emb. | Backb. | Met. | CUB Clar.↑ | Acc.↑ | Spars.↑ | Prec.↑ | Bin-Acc.↑ | SUN Clar.↑ | Acc.↑ | Spars.↑ | Prec.↑ | Bin-Acc.↑ |
|------|--------|------|-------|------|--------|-------|----------|-------|------|--------|-------|----------|
| ViT-B/16 | VLM | $\ell_1$ | 0.36 | 58.7 | .980 | .181 | 88.2 | 0.31 | 55.4 | .810 | .153 | 80.2 |
| | | $\ell_0$ | 0.32 | 76.6 | .838 | .148 | 78.3 | 0.22 | 60.5 | .566 | .101 | 58.8 |
| | | Bern. | 0.40 | 69.7 | .955 | .196 | 87.2 | 0.28 | 56.5 | .726 | .133 | 73.3 |
| | Pred. | $\ell_1$ | 0.63 | 71.2 | .927 | .444 | 89.1 | 0.40 | 51.6 | .816 | .227 | 83.3 |
| | | $\ell_0$ | 0.54 | 73.5 | .821 | .331 | 83.9 | 0.43 | 54.9 | .788 | .258 | 83.2 |
| | | Bern. | 0.69 | 69.6 | .962 | .527 | 90.1 | 0.49 | 52.2 | .863 | .327 | 88.7 |
| ViT-L/14 | VLM | $\ell_1$ | 0.37 | 68.8 | .974 | .175 | 88.2 | 0.32 | 60.2 | .793 | .152 | 79.0 |
| | | $\ell_0$ | 0.33 | 83.2 | .837 | .153 | 78.5 | 0.23 | 65.1 | .557 | .104 | 58.3 |
| | | Bern. | 0.38 | 76.4 | .956 | .182 | 87.1 | 0.28 | 62.0 | .718 | .131 | 72.6 |
| | Pred. | $\ell_1$ | 0.67 | 77.3 | .941 | .474 | 89.6 | 0.41 | 57.3 | .825 | .232 | 84.1 |
| | | $\ell_0$ | 0.56 | 80.0 | .843 | .339 | 84.8 | 0.45 | 59.7 | .801 | .270 | 84.2 |
| | | Bern. | **0.72** | 77.5 | .966 | **.538** | 90.2 | **0.51** | 56.4 | .874 | **.345** | 89.5 |

**Qualitative Analysis.** In Fig. 3, we visualize the selected concepts for two representative models on two examples from the CUB dataset: the model achieving the highest Clarity (Predictor-based Bernoulli) and the model achieving the highest classification accuracy (VLM-based $\ell_0$), as reported in Table 2. We observe that the high-Clarity model selects a compact set of concepts with high precision, whereas the high-accuracy model activates a large number of concepts that do not align with the ground-truth annotations, further showcasing the need for a more in-depth evaluation of concept-based models with respect to interpretability. Further qualitative results are provided in the appendix.

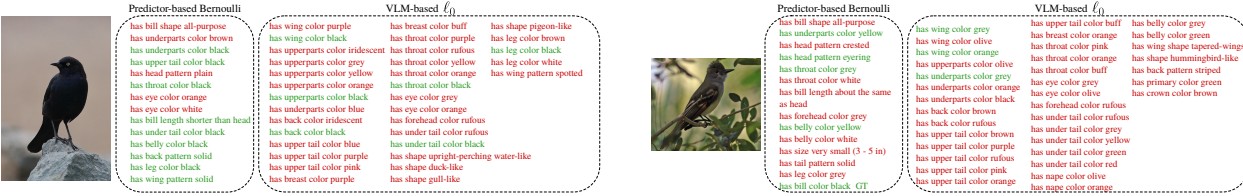

Figure 3: Visualization of selected concepts for two examples from the CUB dataset by selecting the method with the best Clarity (predictor-based Bernoulli) versus the one with the best performance (VLM-based $\ell_0$). Green highlighting denotes concepts that are present in the ground truth. **Left:** *Brewer Blackbird* example with 28 ground-truth active concepts. **Right:** *Great Crested Flycatcher* example with 30 ground-truth active concepts. Similar trends are observed in both cases, with the Bernoulli method selecting fewer, more precise concepts, and the $\ell_0$ method selecting a larger set with lower precision.

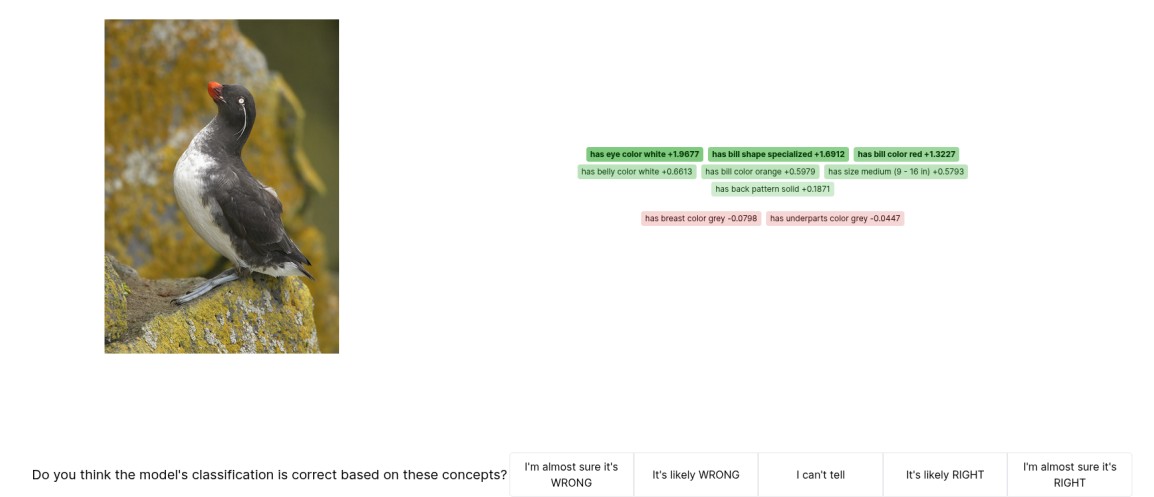

Figure 4: **Human Study Example.** Representative interface. Participants judged model correctness based only on the displayed concepts, which are color-coded by their contribution to the final classification (green for support, red for opposition).

**Human Study.** To evaluate the alignment between our proposed Clarity metric and human perception, we conducted a controlled user study with 50 participants. Each individual was presented with 30 images, sampled from models spanning a diverse spectrum of Clarity scores. For each instance, participants were shown the inferred active concepts for this image and tasked with judging whether the model was able to correctly classify it, based solely on this information; a representative example of the study interface is illustrated in Fig. 4, where the number next to each concept represents its contribution towards the classification and the color signifies if this contributions supports (green) or opposes (red) the classification. Responses were collected on a 5-point Likert scale ranging from *"I'm almost sure it's WRONG"* to *"I'm almost sure it's RIGHT"*. These judgments were subsequently mapped to a numerical scale $[-2, 2]$, allowing us to define **Trust** as the mean score assigned to each method across all participants and samples.

Our results in Table 3 show a strong alignment between human trust and the proposed Clarity metric. In particular, Clarity exhibits the highest correlation with human judgments ($r = 0.9749$), outperforming individual components such as Precision ($r = 0.9404$), as well as common aggregate measures including the Arithmetic Mean ($r = 0.9124$) and the Geometric Mean ($r = 0.9673$). Notably, sparsity and accuracy alone are poor predictors of human trust, with sparsity showing only moderate correlation and accuracy even exhibiting a negative trend. To explicitly disentangle whether human trust scores were driven by individual factors, such as the visual density of the interface (Sparsity), we conducted a post-hoc partial regression analysis. We fitted a multiple linear regression model using standardized variables to predict aggregate user trust. Controlling for visual density reveals that Clarity is the sole meaningful driver of trust

($\beta = 0.9741, p = 0.001$), while Sparsity retains no statistically detectable predictive power ($\beta = 0.0012, p = 0.993$; adjusted $R^2 = 0.931$). Variance Inflation Factor (VIF) analysis confirms that collinearity between predictors does not distort these estimates (VIF = 1.72 for both variables). This suggests that human trust is fundamentally anchored in the joint interplay of sparsity, precision, and accuracy as captured by Clarity rather than any single component baseline, suggesting an architectural alignment of the non-compensatory Clarity metric with human perception. To further assess the robustness of this relationship, we performed

Table 3: Comprehensive Analysis of Evaluation Metrics and Trust.

(a) Correlation with human trust scores

| Metric | Pearson r | p-value |
|---|---|---|
| Clarity | 0.9749 | 0.0001 |
| Sparsity | 0.6322 | 0.0926 |
| Accuracy | -0.4683 | 0.2419 |
| Precision | 0.9404 | 0.0005 |
| Arithmetic Mean | 0.9124 | 0.0016 |
| Geometric Mean | 0.9673 | 0.0001 |

(b) Sensitivity analysis of Clarity weights

| W. Spars. | W. Prec. | W. Acc | Corr |
|---|---|---|---|
| 0.1 | 0.4 | 1.0 | 0.9833 |
| 0.1 | 0.3 | 0.8 | 0.9833 |
| 0.1 | 0.3 | 0.7 | 0.9831 |
| 0.1 | 0.4 | 0.9 | 0.9831 |
| 0.2 | 0.4 | 1.0 | 0.9831 |
| 1.0 | 1.0 | 1.0 | 0.9749 |

a sensitivity analysis by varying the relative weights assigned to sparsity, precision, and accuracy in the Clarity formulation. Across a wide range of weight configurations, the correlation with human trust remains consistently high (up to $r = 0.9833$), indicating that the observed alignment is not dependent on a particular parameter setting but is instead an intrinsic property of the metric.

Beyond quantitative alignment, these results provide insight into *why* Clarity accurately reflects human perception. While the trade-off between predictive accuracy and interpretability is well-documented, our framework makes this gap explicitly measurable in sparsity-aware CBMs. We find that both predictor-based and VLM-based models systematically sacrifice concept precision to maintain task accuracy and sparsity, effectively prioritizing discriminative signals over semantic alignment.

The human study reveals a critical behavioral consequence of this prioritization. High-Clarity models foster trust because their explanations provide coherent, semantically aligned signals that match user intuition. In contrast, low-Clarity models induce what we term *epistemic noise*: a state in which users cannot reliably judge model correctness because the explanation no longer correlates with the model's internal logic. In this regime, interpretability fails not due to a lack of information, but because the information provided is misleading or semantically inconsistent. By unifying sparsity, precision, and accuracy under the metric of Clarity, we establish a formal baseline for *representational integrity*, enabling a more rigorous assessment of how models navigate the tension between performance and transparency.

**Other Architectures.** To move beyond our integrated, per-instance sparsification framework, we extend our analysis to a broader set of state-of-the-art Concept Bottleneck Model (CBM) variants where sparsity is applied globally, i.e., at the parameter or representation level rather than conditioned on individual inputs. First, we consider Post-Hoc CBMs (Yuksekgonul et al., 2022), Label-Free CBMs (Oikarinen et al., 2023), and Sparse-CBMs (Semenov et al., 2024), which represent widely used paradigms in the literature. For all methods, we adhere as closely as possible to their original formulations, training objectives, and default hyperparameters, while also testing different configurations (details are provided in the Appendix). The main modification we introduce is the substitution of the original concept sets with the ground-truth CUB attributes. This adjustment is necessary to enable a consistent and objective evaluation of concept *precision*, which forms a core component of our Clarity metric. We also evaluate a TopK-CBM variant to investigate a fixed-sparsity scenario. Unlike our amortized framework, which learns optimal per-instance masks, TopK-CBM enforces a rigid bottleneck by retaining only a fixed number ($K$) of the most active concepts per-example.

Our results, depicted in Table 4, reveal a consistent pattern across globally-sparse architectures: while several methods achieve strong classification performance, they frequently do so at the expense of concept precision. These baselines were originally developed for large automatically discovered concept vocabularies, rather than bounded human-annotated sets; thus, they seek a different trade-off than our setting, i.e., sacrificing

Table 4: Comparison of Clarity and performance metrics across different CBM architectures. Here, we show the highest achieved clarity for each method. More extensive results can be found in the Appendix.

| Method | Clarity ↑ | Accuracy ↑ | Sparsity ↑ | Precision ↑ |
|---|---|---|---|---|
| *Baseline* | | | | |
| Standard-CBM | 0.6345 | 61.69 | .9359 | **0.6531** |
| *Global/SOTA Baselines* | | | | |
| Post-Hoc CBM (Yuksekgonul et al., 2022) | 0.2463 | 55.50 | .8320 | 0.1090 |
| Label-Free CBM (Oikarinen et al., 2023) | 0.3540 | 70.10 | .962 | 0.1690 |
| Sparse-CBM (Semenov et al., 2024) | 0.2127 | 34.16 | .8380 | 0.1001 |
| *Fixed Sparsity (Top-K)* | | | | |
| VLM-based TopK-32 | 0.2313 | 43.16 | .9000 | 0.1713 |
| Predictor-based TopK-32 | 0.5672 | 60.72 | .9000 | 0.5473 |
| *Amortized (Ours)* | | | | |
| VLM-based Bernoulli | 0.3830 | 76.35 | .9560 | 0.1825 |
| Predictor-based Bernoulli | **0.7170** | **77.48** | **.9660** | 0.5377 |

concept precision in favour of large-scale concepts sets. Consequently, this setting may not fully reflect the operating regime for which some of these approaches were originally designed.

Nevertheless, low Precision cannot be attributed solely to the reduced vocabulary size. In particular, methods that promote class-discriminative sparsity may select concepts that are predictive of the target class without being semantically aligned with the ground-truth attributes of a specific instance. From the perspective of Clarity, such behavior reflects a decoupling between predictive utility and semantic faithfulness, which is precisely the phenomenon the metric is intended to quantify. This behavior mirrors our observations in VLM-based CBMs, suggesting that the issue may not just be architectural but may instead arise whenever semantic grounding is weak or absent. More broadly, these experiments potentially highlight a fundamental limitation of global sparsity mechanisms: they optimize representations for population-level discriminative power without accounting for instance-specific semantic consistency. Consequently, while such solutions may perform well on average, they often fail to produce faithful explanations at the individual level. In contrast, our amortized framework explicitly models this instance-level dependency, enabling more granular control over the *sparsity–precision–accuracy* trade-off. We further observe that Top-K models exhibit only modest Clarity; as $K$ increases, the added architectural flexibility permits the inclusion of irrelevant concepts, leading to a sharp decline in precision and diminishing returns in interpretability. More extensive results and analysis are provided in the Appendix.

Overall, across diverse architectures and training paradigms, we consistently observe that high predictive performance and even high sparsity does not guarantee interpretable behavior, and that concept precision is systematically sacrificed in the absence of explicit constraints. This further supports the need for a unified diagnostic metric such as Clarity, which exposes these failure modes and enables principled model selection.

**Clarity Beyond Sparsity.** While our primary formulation of Clarity incorporates sparsity, precision, and accuracy, the metric generalizes naturally to non-sparse settings. In such cases, where sparsity is not a primary design objective, Clarity reduces to the harmonic mean of *Precision* and *Accuracy*. This simplified form preserves the core intuition of our framework: penalizing models that achieve strong predictive performance at the expense of semantic correctness. Empirically, we observe that sparsity and precision are highly correlated in our experimental setting, leading to consistent model rankings regardless of whether the sparsity term is included. However, this does not render sparsity redundant; rather, it reflects a regime where sparsity acts as an implicit regularizer that promotes precision by pruning irrelevant concept activations. In general, while higher sparsity does not strictly guarantee higher precision, it remains essential for limiting cognitive load and ensuring the interpretability of the bottleneck. Our human evaluation confirms this versatility, yielding comparable trust correlations for both the three-component and two-component formulations, i.e., $r = 0.9749$. Ultimately, this extension highlights the utility of Clarity as a unifying metric: whether in

sparse or dense regimes, it provides a principled basis for assessing the trade-off between task performance and semantic fidelity.

## 5 Limitations and Future Work

A primary limitation of our framework is its reliance on ground-truth concept annotations to calculate *Clarity*. However, we contend that such rigorous evaluation is a vital prerequisite for the development of truly interpretable architectures. These controlled settings offer a robust foundation, serving as a necessary springboard for scaling this analysis to large-scale domains where ground-truth annotations may be sparse or noisy. More broadly, we emphasize that without access to semantically verified concepts, it is difficult to reliably distinguish whether a model is semantically grounded thus yielding accurate concept representations, reinforcing the importance of such benchmarks as the bedrock of interpretability research. By validating Clarity in this principled environment, we establish a foundation that can be extended to more unconstrained, open-world applications.

A significant challenge in extending Clarity to open-vocabulary Concept Bottleneck Models (CBMs) is the absence of per-instance ground-truth annotations, which makes the Precision component difficult to compute directly. A natural direction is therefore to *approximate* Precision through a combination of proxy signals that capture semantic alignment at different levels. For instance, one could leverage external knowledge sources to assess the plausibility and consistency of activated concept combinations, or exploit visual grounding signals to verify instance-level evidence. Furthermore, counterfactual interventions, e.g., using generative models to modify or suppress specific visual features, could provide a more direct test of concept faithfulness by evaluating whether activations respond predictably to controlled changes in the input. Additional indicators, such as cross-modal agreement or the stability of activations under perturbations, may further provide indirect evidence of alignment.

Crucially, the non-compensatory nature of the Clarity metric, by virtue of the harmonic mean, ensures that even approximate or noisy precision signals can meaningfully penalize models that prioritize task performance over semantic consistency. While these proxies are inherently imperfect, they suggest a principled path toward evaluating interpretability in open-world scenarios where ground-truth annotations are unavailable, and we consider this a promising direction for future work.

*Clarity* is intentionally designed as a minimal, assumption-light diagnostic rather than a tunable aggregation scheme. While it combines *Sparsity*, *Precision*, and *Accuracy* through a harmonic mean with equal weights, our sensitivity analysis demonstrates that its strong alignment with human trust is robust across a wide range of weight configurations. This suggests that the observed behavior is not an artifact of specific hyperparameter choices, but rather reflects an intrinsic relationship between these three dimensions.

Importantly, our goal is not to frame interpretability as a standard multi-objective optimization problem where trade-offs are freely navigated along a Pareto frontier. Instead, we argue that sparsity, precision, and accuracy define a *conjunctive validity criterion*: a model that fails in any one dimension, for instance, by achieving high accuracy and sparsity via semantically incorrect concepts, is effectively uninterpretable. The harmonic mean formulation formalizes this "weakest-link" principle, ensuring that a failure in any single component catastrophically impacts the total score. In this sense, *Clarity* represents a baseline sanity check for representational integrity, rather than a substitute for application-specific qualitative analyses.

Beyond evaluation, this conjunctive, non-compensatory view of interpretability suggests a natural extension of the Clarity framework: rather than measuring semantic misalignment post-hoc, one could incorporate precision-aware signals directly into the training objective. By leveraging the *non-compensatory* principles of Clarity as a training-time regularizer, ground-truth or approximate concept supervision could penalize configurations that achieve high accuracy through semantically incorrect activations. We leave this direction for future work.

## 6 Conclusions

In this work, we presented a principled framework for the systematic evaluation of sparsity-aware Concept Bottleneck Models. We introduced novel amortized formulations for per-instance attribute selection using $\ell_0$ and $\ell_1$ constraints, alongside *Clarity*: a unified metric that formalizes the critical interplay between sparsity, semantic precision, and downstream performance. Our experiments demonstrate that models with near-identical accuracy can exhibit drastically different interpretability profiles, uncovering a fundamental trade-off between architectural *flexibility* and semantic *precision*. These findings underscore that accuracy-centric evaluations are insufficient for CBMs; instead, objective diagnostics like *Clarity* are essential for identifying models that provide semantically faithful and trustworthy explanations.

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

## A  Background on sparse representations

Sparsity-aware learning has a long history, spanning from early signal processing and compressed sensing to modern deep architectures. While a variety of methods exist, they all share a common principle: an underlying *sparse code* is assumed to exist that can adequately represent the data; this assumption is explicitly enforced during learning to enhance interpretability, efficiency, and generalization. In the context of machine learning, one of the most common approaches towards sparsity is the $\ell_1$ regularization (Theodoridis, 2015), where the optimization process is given by:

$$\min_{\boldsymbol{\theta}} \frac{1}{N} \sum_{i=1}^{N} \mathcal{L}^{\text{task}}(f(\boldsymbol{x}_i; \boldsymbol{\theta}), y_i) + \lambda ||\boldsymbol{\theta}||_1 \tag{16}$$

where $||\boldsymbol{\theta}||_1 = \sum_{j=1}^{M} |\theta_j|$ is the $\ell_1$ norm of $\boldsymbol{\theta}$ and $\lambda \geq 0$ is a tunable parameter controlling the penalty contribution. A key property of the $\ell_1$ constraint is that it promotes sparse solutions, typically computed via coordinate descent.

The $\ell_1$ norm is simple, effective and easy to implement, which has contributed to its widespread adoption in modern ML. Nevertheless, more expressive priors exist that can further enhance interpretability. For instance, the $\ell_0$ "norm" can enforce sparsity by directly counting non-zero parameters, potentially leading to even more precise sparse representations. However, the combinatorial nature of the $\ell_0$ definition renders the optimization problem intractable. To bypass this issue, (Louizos et al., 2018) relax the discrete nature of the optimization, introducing a construction based on the *Hard Concrete* distribution.

The empirical loss minimization with $\ell_0$ penalty reads:

$$\min_{\boldsymbol{\theta}} \frac{1}{N} \sum_{i=1}^{N} \left( \mathcal{L}^{\text{task}}(f(\boldsymbol{x}_i; \boldsymbol{\theta}, y_i)) + \lambda ||\boldsymbol{\theta}||_0, \qquad \text{where } ||\boldsymbol{\theta}||_0 = \sum_{j=1}^{M} \mathbb{I}[\theta_j \neq 0] \tag{17}$$

We can reparameterize the parameter vector $\boldsymbol{\theta}$, s.t.:

$$\theta_j = \tilde{\theta}_j z_j,$$
$$\text{where } z_j \in \{0, 1\}, \quad \tilde{\theta}_j \neq 0, \quad ||\boldsymbol{\theta}||_0 = \sum_{j=1}^{M} z_j \tag{18}$$

where $z_j, j = 1, \ldots, M$ constitute auxiliary binary latent variables, that denote if each parameter is present or absent; thus, the $\ell_0$ term now acts as a counter of the components that are active each time. By imposing an appropriate Bernoulli distribution on the latent variables $\boldsymbol{z}$, such that $z_j \sim \text{Bernoulli}(\pi_j)$, $\forall j$, the empirical loss function can be written as follows (Louizos et al., 2018):

$$\mathbb{E}_{q(\boldsymbol{z}|\boldsymbol{\pi})} \left[ \frac{1}{N} \left( \sum_{i=1}^{N} \mathcal{L}^{\text{task}}(f(\boldsymbol{x}_i; \boldsymbol{z} \cdot \tilde{\boldsymbol{\theta}}, \boldsymbol{y}_i)) \right) \right] + \lambda \sum_{j=1}^{M} \pi_j \tag{19}$$

Again, the discrete nature of $\boldsymbol{z}$ renders gradient-based optimization inefficient, with REINFORCE estimators (Williams, 1992) suffering from high variance, while the Concrete distribution (Maddison et al., 2017; Jang et al., 2017) and the Straight Through estimators suffer from their own drawbacks, i.e., biased gradients and not exact zeroes respectively. Thus, the authors introduce a continuous random variable $\boldsymbol{s}$, governed by a distribution $q(\boldsymbol{s})$ and parameters $\boldsymbol{\phi}$, followed by a hard-sigmoid rectification, yielding the so-called *Hard Concrete* distribution. The construction reads:

$$\boldsymbol{s} \sim q(\boldsymbol{s}|\boldsymbol{\phi}) \tag{20}$$
$$\boldsymbol{z} = \min(\mathbf{1}, \max(\mathbf{0}, \boldsymbol{s})), \tag{21}$$

In practice, the Hard Concrete distribution treats $\boldsymbol{s}$ as a continuous relaxation of the Bernoulli distribution, parameterized by $\phi = (\log \alpha, \beta)$, where $\log \alpha$ is the location and $\beta$ is the temperature, yielding the following

sampling procedure:

$$s = \sigma((\log u - \log(1-u) + \log \alpha)/\beta) \tag{22}$$

$$\bar{s} = s(\zeta - \gamma) + \gamma \tag{23}$$

$$z = \min(1, \max(0, \bar{s})) \tag{24}$$

where $u \sim \mathcal{U}(0,1)$, $\sigma(\cdot) = \text{Sigmoid}(\cdot)$, and $\gamma < 0$ and $\zeta > 1$ can be used to stretch the arising distribution to the $(\gamma, \zeta)$ interval. The final objective function comprises the *error* and the *complexity loss* (Louizos et al., 2018); it is commonly estimated through MC sampling with $L$ samples:

$$\mathcal{L} = \frac{1}{L} \sum_{l=1}^{L} \left( \frac{1}{N} \sum_{i=1}^{N} \mathcal{L}^{\text{task}}(f(\boldsymbol{x}_i; \tilde{\boldsymbol{\theta}} \cdot \boldsymbol{z}^{(l)}), \boldsymbol{y}_i) \right) + \sum_{j=1}^{M} \text{Sigmoid} \left( \log \alpha_j - \beta \log \frac{-\gamma}{\zeta} \right) \tag{25}$$

Direct optimization of the $\ell_0$ norm is combinatorially hard, and methods like Hard Concrete provide continuous relaxations to make it differentiable for gradient-based optimization. An alternative approach is to model sparsity probabilistically using Bernoulli variables, where each parameter is stochastically "switched on/off," enabling principled learning of sparse structures in a fully differentiable framework.

Drawing inspiration from latent feature models (Griffiths & Ghahramani, 2011), we can introduce a set of auxiliary binary latent variables $\boldsymbol{z} \in \{0,1\}^M$, that dictate which components are active at any given time. In analogy to the $\ell_0$ formulation, we can reparameterize the parameter vector $\boldsymbol{\theta}$, such that:

$$\theta_j = \tilde{\theta}_j \cdot z_j, \quad z_j \in \{0,1\} \tag{26}$$

We can now impose an appropriate prior on these latent variables, i.e., a Bernoulli distribution, such that $\boldsymbol{z} \sim \text{Bernoulli}(\tilde{\boldsymbol{\pi}})$ and perform inference on both the parameter vector $\boldsymbol{\theta}$ and the latent variables $\boldsymbol{z}$. Using a posterior mean-field approach $q(\boldsymbol{z}|\pi) = \text{Bernoulli}(\boldsymbol{\pi})$ (Beal, 2003; Wainwright & Jordan, 2008), the Evidence Lower Bound (ELBO) expression reads:

$$\text{ELBO} = \mathbb{E}_{q(\boldsymbol{z}|\boldsymbol{\pi})} \left[ \frac{1}{N} \sum_{i=1}^{N} \mathcal{L}(f(\boldsymbol{x}_i; \tilde{\boldsymbol{\theta}} \cdot \boldsymbol{z}), y_i) \right] - KL[q(\boldsymbol{z}|\boldsymbol{\pi})||p(\boldsymbol{z}|\tilde{\boldsymbol{\pi}})]. \tag{27}$$

By placing a suitable sparsity-inducing prior on $\boldsymbol{z}$, i.e., $\boldsymbol{z} \sim \text{Bernoulli}(\tilde{\boldsymbol{\pi}})$, we encourage only a subset of parameters to be active while performing joint inference over $\boldsymbol{\theta}$ and $\boldsymbol{z}$.

Building on these definitions of the three sparsity-aware formulations, we now turn to their use in designing per-example sparsity-aware concept selection mechanisms.

### A.1 Regularization Terms

For the $\ell_1$-based method, we impose the regularization penalty on the activations $\boldsymbol{z}_i$ themselves, since the goal is to penalize the concept activations rather than the amortization matrix. For $\ell_0$-based sparsity, the penalty corresponds to the second term in Eq. 25, where $\log \alpha = E_I(\boldsymbol{X})\boldsymbol{W}_s^\top \in \mathbb{R}^{N \times M}$, s.t.,:

$$\mathcal{L}_{\ell_0}^{\text{penalty}} = \frac{1}{N} \sum_{i,m}^{N,M} \sigma \left( E_I(\boldsymbol{X})\boldsymbol{W}_s^\top - \beta \log \frac{-\gamma}{\zeta} \right) \tag{28}$$

Finally, for the Bernoulli-based formulation, and since Bernoulli is not amenable to the reparameterization trick, we consider its relaxation based on the Concrete distribution (Maddison et al., 2017; Jang et al., 2017) to draw samples from the approximate posterior during training; for the "penalty", we compute the KL divergence between a fixed sparsity-inducing prior and the Bernoulli posterior, thus optimizing the loss provided in Eq. 27.

# B  Experimental Details.

For our experiments, we set the Bernoulli prior to a very small value, $10^{-4}$, to strongly enforce sparsity. For methods that use the continuous relaxation of the Bernoulli distribution (i.e., Bernoulli- and $\ell_0$-based methods), we use a temperature of 0.1, while we set $\gamma = -0.1$ and $\zeta = 1.1$ for the HardConcrete distribution following (Louizos et al., 2018).

For both Bernoulli and Hard Concrete based experiments, we compute the loss using $L = 1$ sample. We did not observe any impact when using more samples.

We explored a wide range of relevant hyperparameters for all methods: learning rates $[10^{-2}, 5 \cdot 10^{-3}, 10^{-3}, 5 \cdot 10^{-4}, 10^{-4}]$, $\lambda$ for $\ell_1$ regularization $[10^{-5}, 5 \cdot 10^{-5}, 10^{-6}, 5 \cdot 10^{-6}, 10^{-7}]$, and $\lambda$ for $\ell_0$-related sparsity $[10^{-2}, 5 \cdot 10^{-2}, 10^{-1}, 5 \cdot 10^{-1}, 1]$. Additionally, we evaluated a wide range of thresholds: $[10^{-4}, 10^{-3}, 5 \cdot 10^{-3}, 10^{-2}, 2 \cdot 10^{-2}, 3 \cdot 10^{-2}, 4 \cdot 10^{-2}, 5 \cdot 10^{-2}, 7 \cdot 10^{-2}, 10^{-1}, 2 \cdot 10^{-1}, 3 \cdot 10^{-1}, 5 \cdot 10^{-1}, 6 \cdot 10^{-1}, 7 \cdot 10^{-1}, 8 \cdot 10^{-1}, 9 \cdot 10^{-1}]$.

For the image/text encoders, we use ViT-B/16 and ViT-L/14, which produce embeddings of dimensionality 512 and 768, respectively. All models were trained using the Adam optimizer without any complex annealing schedules, on a single NVIDIA A6000 GPU with 48GB of VRAM.

**Variability Analysis.**   To assess the impact of stochastic optimisation noise, we conducted five independent runs under different random seeds for VLM-based Bernoulli models using the CUB dataset, yielding low standard deviations ($\sigma_{\text{acc}} = 0.60\%$, $\sigma_{\text{sparsity}} = 0.0010$, $\sigma_{\text{prec}} = 0.0078$, and $\sigma_{\text{clarity}} = 0.0107$) for ViT-B/16, and ($\sigma_{\text{acc}} = 0.23\%$, $\sigma_{\text{sparsity}} = 0.0005$, $\sigma_{\text{prec}} = 0.0053$, and $\sigma_{\text{clarity}} = 0.0077$) for ViT-L/14, suggesting that the observed patterns and behaviour are not an artifact of optimization variance and that the proposed thresholding strategy is highly robust against such noise.

## B.1  Concrete Relaxation

Let $\boldsymbol{\pi_i}$ denote the probabilities of $q(\boldsymbol{z}_i)$ for $i = 1, \ldots, N$. We can obtain reparameterized samples $\hat{\boldsymbol{z}_i} \in (0,1)^M$ from the continuous relaxation as

$$\hat{\boldsymbol{z}_i} = \frac{1}{1 + \exp\left(-(\log \boldsymbol{\pi_i} + L)/\beta\right)}, \tag{29}$$

where $L \in \mathbb{R}$ is a sample from the Logistic distribution, defined as

$$L = \log U - \log(1 - U), \quad U \sim \text{Uniform}(0, 1), \tag{30}$$

and $\beta$ is the *temperature* parameter controlling the smoothness of the relaxation: higher $\beta$ values produce samples closer to uniform, while lower values yield samples closer to binary. In all experiments, we set $\beta = 0.1$. During inference, the discrete Bernoulli distribution can be used to directly draw binary indicators.

## B.2  Human Study Details.

Participants were recruited through Prolific and performed the task via a Gorilla.sc interface. No domain-specific expertise was required for participation. To ensure fair remuneration, participants were compensated at a rate of €10.35 per hour, exceeding the platform's minimum pay requirements at the time of the study. We randomly selected 100 examples from the CUB validation set and for all our baseline models, we computed the inferred concepts and their contributions. This led to a pool of 800 images-concepts pairs; each participant evaluated a randomized subset of 30 image-model pairs to mitigate ordering effects.

Instructions were standardized and included examples to ensure consistent interpretation of the task. To quantify user trust, we map the 5-point Likert scale ratings from our study to numerical values ranging from -2 (*"I'm almost sure it's WRONG"*) to 2 (*"I'm almost sure it's RIGHT"*). We then aggregate these scores by method to compute a mean trust rating for each model configuration.

Overall, predictor-based models achieve substantially higher trust scores than VLM-based models. The predictor-based approach attains a mean trust score of 0.789 (95% CI: $[0.703, 0.875]$), whereas the VLM-based model yields a mean of $-0.033$ (95% CI: $[-0.134, 0.067]$). Notably, the confidence interval for the

VLM-based model includes zero, indicating that its expected trust is not reliably positive, while the pred-based model shows consistently strong and positive trust across samples.

These results further support the alignment between the proposed Clarity metric and human judgments, suggesting that higher Clarity is associated with increased perceived trustworthiness.

Table 5: Comparison of trust scores and Clarity across methods (mean $\pm$ standard deviation). Pred-based models achieve significantly higher trust scores than VLM-based models ($t = 12.21, p < 10^{-32}$), with a medium-to-large effect size (Cohen's $d = 0.64$).

| Methods | Trust Score | Clarity |
|---------|-------------|---------|
| Pred-based | $0.789 \pm 1.193$ | $0.561 \pm 0.071$ |
| VLM-based | $-0.033 \pm 1.378$ | $0.249 \pm 0.034$ |

### B.3 Experimental Details & Results: Other Architectures

**Training details.** All models were trained following their respective original implementations and default configurations; this includes the optimizers and the hyperparameters. The main change pertains to the use of the ground truth CUB attribute names for the concept set. We additionally explored different configurations for Post-Hoc CBMs and Label-Free CBMs as follows:

- Post-Hoc CBMs: the two main hyperparameters to tune are the regularization strength of the Elastic Net ($\lambda$) and the balancing term between the $L_1$ and $L_2$ penalties ($\alpha$). We considered three different levels of regularization strength, $\lambda \in [10^{-5}, 10^{-4}, 10^{-3}]$, and two configurations for the balancing term: the default of the original implementation ($\alpha = 0.99$) and the default for the standard scikit-learn package ($\alpha = 0.15$). Even though we observed significantly different behaviors in terms of bottleneck sparsity and downstream classification accuracy, the concept precision remained largely stagnant around 10–11

- Label-Free CBMs: We investigated the optimization depth by sweeping projection layer steps (1000 vs. 5000) and SAGA solver iterations (1000 vs. 5000). We discovered that the default hyperparameters (1000 steps) did not always allow for proper convergence harming the precision and clarity of the model. We also tested the regularization strength and balancing term as in Post-Hoc CBMs, without nevertheless noticing any significant changes.

For all methods, we consider the ViT-L/14 backbone for the embeddings.

**Baseline Results.** In Table 6, the results for a "standard" CBM formulation are depicted. In this instance, we first train the predictor using the BCE loss; then, we train a classifier with different thresholds to measure the classification performance and compute all the required components. The threshold is applied to the concept scores after the sigmoid transformation, thus it ranges from 1e-4 to 9e-1.

**Top-K Results.** To further contextualize the performance of our amortized sparsity strategies, we evaluate TopK-CBM, a hard-thresholding approach that retains only the $K$ concepts based on their respective concept scores (either Predictor- or VLM-based). This method serves as a non-probabilistic, fixed sparsity baseline. To explore a wide spectrum of the flexibility-interpretability trade-off, we vary the parameter $K$ across values that reflect both the average ground-truth concept density and more flexible regimes. Specifically, we set $K \in \{4, 8, 16, 32, 64\}$ for the CUB dataset. Consistent with the rest of our framework, we use the ground-truth attribute set to ensure that the measured *precision* remains semantically grounded. The obtained results are depicted in Table 7. We observe that TopK models exhibit a characteristic trade-off as K increases. While larger values of K improve classification accuracy, they lead to a systematic decrease in concept precision, reflecting increased flexibility in concept selection. As a result, Clarity peaks at intermediate values of K and degrades in more flexible regimes. This behavior highlights the limitations of fixed sparsity strategies, which cannot adapt to instance-specific concept relevance.

Table 6: Performance of a **Predictor-based CBM**, similar to (Koh et al., 2020), across representative threshold regimes.

| Threshold | Predictor-based CBM | | | |
|---|---|---|---|---|
| | Clarity ↑ | Accuracy ↑ | Sparsity ↑ | Precision ↑ |
| 1e-4 | 0.1868 | 81.09 | 0.0437 | 0.1056 |
| 1e-3 | 0.2178 | 81.28 | 0.2014 | 0.1257 |
| 1e-2 | 0.3065 | 81.07 | 0.4806 | 0.1890 |
| 2e-2 | 0.3488 | 81.11 | 0.5659 | 0.2222 |
| 5e-2 | 0.4201 | 80.52 | 0.6763 | 0.2842 |
| 1e-1 | 0.4875 | 79.26 | 0.7584 | 0.3520 |
| 3e-1 | 0.6081 | 72.38 | 0.8831 | 0.5243 |
| 5e-1 | **0.6345** | 61.69 | 0.9359 | 0.6531 |
| 7e-1 | 0.5818 | 46.87 | 0.9675 | 0.7667 |
| 9e-1 | 0.2616 | 15.37 | 0.9919 | **0.8779** |

Table 7: Comparative analysis of Predictor-based vs. VLM-based models across fixed sparsity (Top-K) regimes.

| | Predictor-based Models | | | | VLM-based Models | | | |
|---|---|---|---|---|---|---|---|---|
| | Clarity ↑ | Accuracy ↑ | Sparsity ↑ | Precision ↑ | Clarity ↑ | Accuracy ↑ | Sparsity ↑ | Precision ↑ |
| TopK-4 | 0.3135 | 22.60 | 0.987 | **0.8027** | 0.1949 | 21.65 | 0.987 | 0.2158 |
| TopK-8 | 0.4392 | 34.12 | 0.974 | 0.7538 | 0.2206 | 28.20 | 0.974 | 0.2118 |
| TopK-16 | 0.5588 | 47.82 | 0.949 | 0.6755 | 0.2249 | 34.20 | 0.949 | 0.1919 |
| TopK-32 | **0.5672** | 60.72 | 0.900 | 0.5473 | 0.2313 | 43.16 | 0.900 | 0.1713 |
| TopK-64 | 0.5168 | **70.80** | 0.800 | 0.3834 | 0.2137 | 50.62 | 0.800 | 0.1406 |

**Global Sparsity Models Results.** To ensure a comprehensive evaluation, we include comparisons with models that utilize global sparsity constraints, specifically Post-hoc CBMs (Yuksekgonul et al., 2022), Label-free CBMs (Oikarinen et al., 2023) and Sparse-CBMs (Semenov et al., 2024). Unlike our amortized strategies which learn per-instance masks, these methods typically impose sparsity on the weights of the final linear layer (e.g., via $\ell_1$ regularization or the GLM-saga sparsification process). To evaluate these models within our Clarity framework, we map global weight sparsity to local explanations by thresholding the contribution of each concept to the predicted class. Specifically, a concept $j$ is considered active for an instance $x$ if its weighted contribution $|w_{yj} \cdot \hat{c}_j(x)|$ exceeds a threshold $\tau$, where $w_{yj}$ is the classifier weight for the predicted class $y$ and $\hat{c}_j(x)$ is the score of concept $j$ for the instance $x$. This conversion allows us to calculate per-example *precision* and *sparsity* for globally-regularized models, facilitating a direct comparison of their semantic alignment against instance-aware sparsity methods.

Table 8: Unified comparison of CBM variants across sparsity regimes. Values represent representative operating points along each method's threshold sweep.

| Regime | PostHoc CBM | | | | Label-Free CBM | | | | Sparse-CBMs | | | |
|---|---|---|---|---|---|---|---|---|---|---|---|---|
| | Clar.↑ | Acc.↑ | Spar.↑ | Prec.↑ | Clar.↑ | Acc.↑ | Spar.↑ | Prec.↑ | Clar.↑ | Acc.↑ | Spar.↑ | Prec.↑ |
| Low sparsity | 0.2357 | 56.37 | 0.7652 | 0.1037 | 0.3308 | 74.17 | 0.8970 | 0.1510 | 0.0282 | 63.73 | 0.0105 | 0.1009 |
| Mid regime | 0.2400 | 56.39 | 0.7926 | 0.1057 | 0.3371 | 74.00 | 0.9112 | 0.1510 | 0.2001 | 24.90 | 0.9294 | 0.1010 |
| Transition | 0.2432 | 56.42 | 0.8083 | 0.1072 | 0.3456 | 73.48 | 0.9390 | 0.1600 | 0.1615 | 12.97 | 0.9758 | 0.1016 |
| High sparsity | 0.2366 | 49.64 | 0.9009 | 0.1047 | 0.3409 | 39.05 | 0.9833 | 0.1905 | 0.0151 | 0.66 | 0.9994 | 0.0221 |

## C    Additional Visualizations

### C.1    Inferred Concepts

To supplement our quantitative findings, we provide extended visualizations in Figures 5 and 6, comparing the concept activations across different models. These visualizations highlight a recurring failure mode in high-flexibility models (e.g., VLM-based $\ell_0$): to maintain high downstream accuracy, these models often "hallucinate" attributes, selecting concepts that are semantically absent from the image but statistically correlated with the class label. In contrast, models with high *Clarity* demonstrate superior semantic grounding. As seen in the Belted Kingfisher and Chaparral examples, high-Clarity models activate a sparse, precise subset of concepts that closely matches the human-annotated ground truth. This qualitative evidence reinforces our claim that sparsity and accuracy alone does not guarantee interpretability; rather, the alignment among accuracy, sparsity and the semantic precision of the bottleneck is what determines the trustworthiness of the resulting explanation.

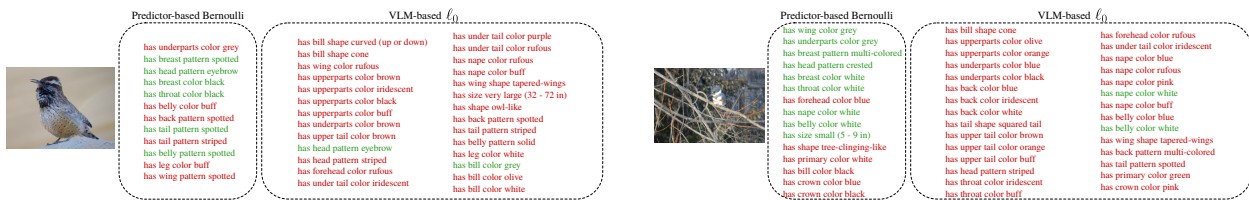

Figure 5: Visualization of selected concepts for two examples from the CUB dataset for the best performing models in terms of Clarity (`Predictor-based Bernoulli`) and classification performance (`VLM-based $\ell_0$`). Green highlighting denotes concepts that are active in the ground truth. **Left:** *Cactus Wren* example with 34 ground-truth active concepts; the Bernoulli-based method selects 12 concepts, 6 of which are correct, while the $\ell_0$ method selects 27 concepts, 2 of which are correct. **Right:** *Belted Kingfisher* example with 28 ground-truth active concepts; similar trends are observed, with the Bernoulli-based method selecting fewer, more precise concepts, and the $\ell_0$ method selecting a larger set with lower precision.

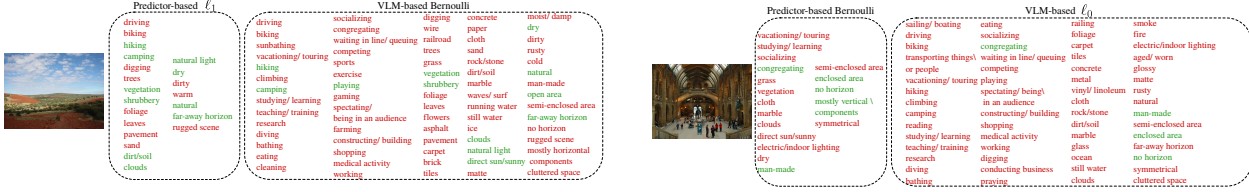

Figure 6: Visualization of selected concepts for two examples from the SUN dataset. Green highlighting denotes concepts that are active in the ground truth. **Left:** *Chaparral* example with 13 ground-truth active concepts; we choose a different set of models from the optimal in terms of Clarity or accuracy, i.e., (`Predictor-based $\ell_1$`) and classification performance (`VLM-based Bernoulli`). The $\ell_1$-based method selects 20 concepts, 10 of which are correct, while the Bernoulli-based method selects 54 concepts, 11 of which are correct. **Right:** *Natural History Museum* example with 5 ground-truth active concepts; here, we choose the best performing models in terms of either Clarity of accuracy as before, i.e., `Predictor-based Bernoulli` and `VLM-based $\ell_0$`. The Bernoulli-based method selects fewer, more precise concepts, and finds the 5 ground truth concepts, while the $\ell_0$ method selects a larger set with lower precision.

### C.2    Human Study Examples

To validate our proposed metric, we provide extended visualizations of the interface presented to participants during our human study (see Figure 7). Each trial presented a model's top-activated concepts, considering a maximum of 70 concepts shown to not overwhelm the users, color-coded by their contribution to the final class prediction (green for supporting, red for opposing). Qualitatively, the high-Clarity models consistently produced "cleaner" profiles where a small number of semantically relevant concepts (e.g., specific plumage

patterns for CUB) dominated the decision. In contrast, high-flexibility, low-Clarity models often presented a "noisy" mix of supporting and opposing concepts that made the final classification appear arbitrary. These visualizations serve to ground our quantitative correlation results, illustrating that the *Clarity* metric accurately penalizes the cognitive load and confusion caused by semantically inconsistent or overly dense concept bottlenecks.

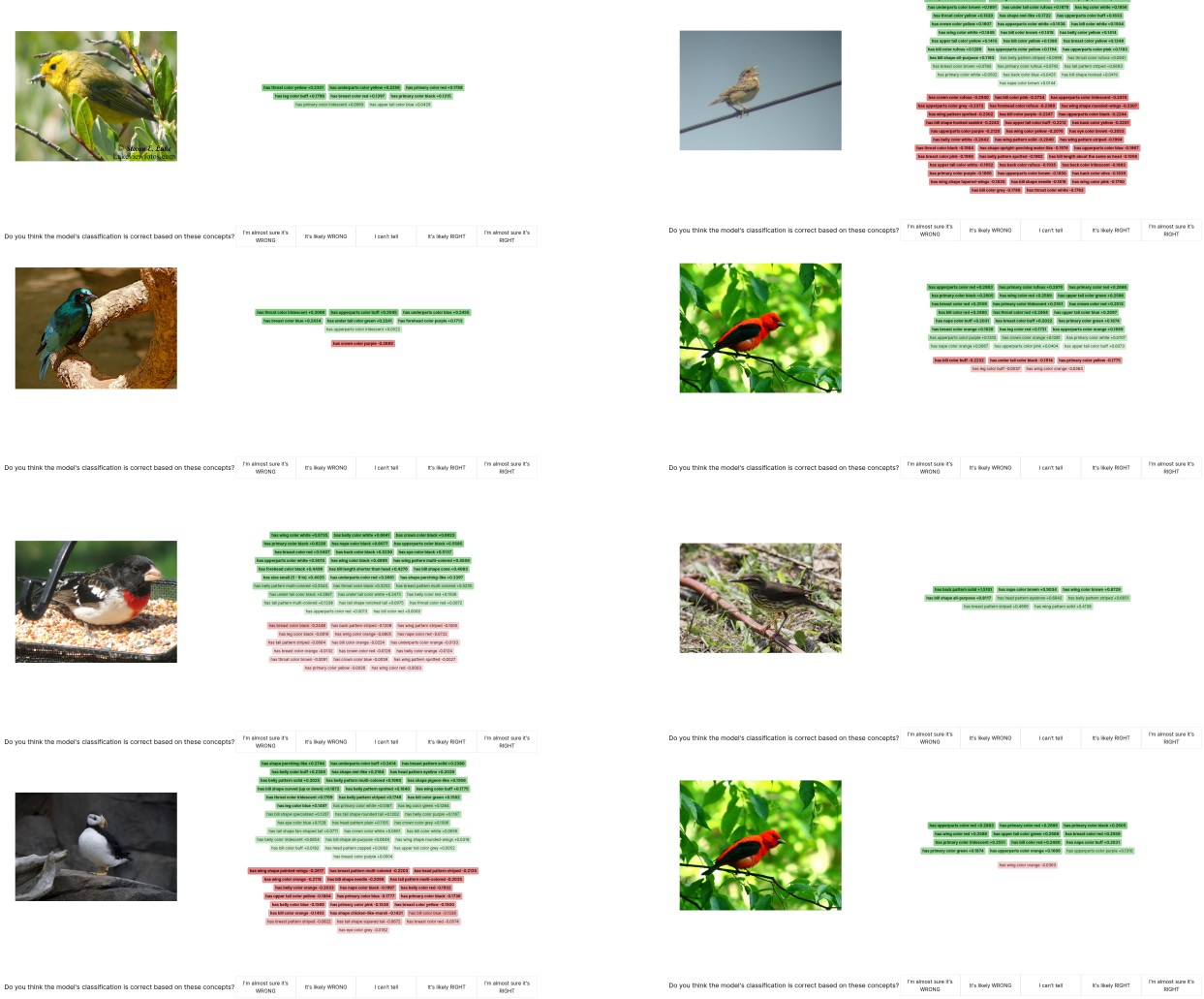

Figure 7: **Human Study Example.** Participants judged model correctness based only on displayed concepts, color-coded by contribution (green = support, red = opposition).

