# OpenReview forum: "Clarity: The Flexibility-Interpretability Trade-Off in Sparsity-aware Concept Bottleneck Models"
_TMLR — Under review for TMLR_

### Review · Reviewer_xNUu · 2026-05-30

**Summary Of Contributions:**

This paper addresses the evaluation of interpretability in Concept Bottleneck Models (CBMs), arguing that standard metrics such as classification accuracy and overall sparsity are insufficient to assess whether a model produces semantically faithful explanations. The main contributions are: the formalization of the flexibility-interpretability trade-off, defined as a model's capacity to maintain high predictive performance by deviating from semantically correct concept representations; the introduction of Clarity, a unified metric defined as the harmonic mean of sparsity, concept precision, and downstream accuracy; a comprehensive empirical evaluation across multiple CBM architectures and sparsity-inducing strategies on the CUB and SUN datasets; and a human study with 50 participants showing that Clarity correlates more strongly with human trust than individual components or standard aggregation schemes.

Key strengths: the core intuition behind Clarity is well-motivated and the harmonic mean formulation elegantly captures the weakest-link property. The human study is a genuine attempt to ground the metric in human perception, which distinguishes this work from purely quantitative interpretability papers. The empirical finding that models with similar accuracy can exhibit dramatically different Clarity levels is an important and practically relevant observation.

Key weaknesses: the three components of Clarity are never formally defined; the comparison with global-sparsity baselines is conducted in a setup that may disadvantage those methods and the human study cannot fully separate the effect of semantic quality from the effect of visual density of the interface. (see below for better motivations of weaknesses)

**Audience:**

Yes

**Audience Explanation:**

The flexibility-interpretability trade-off identified in this paper is a genuine and underexplored problem in the CBM literature. The observation that models with near-identical accuracy can exhibit drastically different levels of semantic alignment is practically relevant for anyone deploying concept-based models in settings where interpretability matters. The proposed Clarity metric, despite the definitional gaps noted above, offers a principled starting point for more rigorous evaluation of CBMs. Researchers working on interpretable machine learning, concept-based explanations, and human-AI trust would find the findings of this paper relevant.

**Broader Impact Concerns:**

No specific broader impact concerns.

**Claims And Evidence:**

No

**Claims Explanation:**

The claims are partially supported but not fully convincing due to three issues.

1. The metric Clarity is the central contribution of the paper, yet its three components, Sparsity, Precision, and Accuracy, are never formally defined. Section 4.2 introduces them only qualitatively and jumps directly to the harmonic mean formula. It is never specified how Sparsity is computed (presumably as the fraction of inactive concepts per example, but this is only inferable from a remark in Table 2), how Precision depends on the threshold τ used to determine concept activation. Without these definitions Clarity is not reproducible, which undermines the paper's core claim.

2. The comparison with Post-Hoc CBM, Label-Free CBM and Sparse-CBM in Table 4 is conducted by replacing their original concept sets with the 312 ground-truth CUB attributes. These methods were designed and optimized for large, automatically generated concept sets of thousands of concepts, and their sparsification logic is calibrated on that regime. Evaluating them with a small, highly correlated concept set may artificially disadvantage them regardless of their intrinsic quality. Label-Free CBM achieves Clarity = 0.002 and Precision = 0.0006, a result so extreme that it suggests the method is simply operating outside its intended regime rather than being genuinely inferior. The authors should either discuss this limitation explicitly or provide additional experiments evaluating these methods in their original setup.

3. The human study presents a confound that is not addressed. Looking at Table 2 and Figure 3, models with high Clarity systematically activate few concepts and semantically correct ones, while models with low Clarity systematically activate many concepts and semantically incorrect ones. These two factors, number of active concepts and their semantic quality, are never separated in the study design. It is therefore not possible to exclude that the observed correlation reflects users reacting primarily to the visual density of the interface rather than to the semantic coherence of the concepts.

**Requested Changes:**

1. Provide explicit formal definitions of all three components of Clarity — Sparsity, Precision, and Accuracy — including their dependence on the threshold τ and how τ is chosen across different experimental settings (VLM-based vs Predictor-based, CUB vs SUN). Without these definitions the metric is not reproducible.

2. Discuss more explicitly the limitations of the comparison with Post-Hoc CBM, Label-Free CBM and Sparse-CBM, acknowledging that substituting their original concept sets with the CUB ground-truth attributes may place these methods in a regime for which they were not designed.

3. Discuss whether the correlation between Clarity and human trust reflects the semantic quality of the concepts or simply the number of concepts shown in the interface, for example by reporting the average number of active concepts per model configuration and verifying via partial regression whether Clarity explains additional variance in trust scores after controlling for this factor. Other explanations to disentangle these two effects would also be considered.

---

> ### Author Response · Authors · 2026-06-04
>
> We thank the reviewer for their constructive feedback. We are pleased that the reviewer found the core intuition behind Clarity well-motivated, our formalization of the flexibility-interpretability trade-off genuine and underexplored, and our human validation study a significant differentiator from purely quantitative papers.
>
> The reviewer highlighted three main limitations that prevented an initial endorsement of our evidence. We have addressed each of these concerns through additions, structural contextualization of the baselines, and a statistical analysis of the user study data.
>
> **R1: Component Definitions and Threshold Selection Strategy**
>
> We agree that formalizing these terms is vital for completeness and reproducibility. We revised Section 3.2 of the manuscript to provide explicit mathematical definitions of Sparsity, Precision, and Accuracy, including their dependence on the activation threshold τ. We also added a dedicated paragraph in Section 4 describing threshold selection across all experimental settings.
>
> **R2: Contextualizing Other Sparsity Baselines**
>
> We appreciate the reviewer's vital observation. Indeed, these baselines were originally designed to operate over large, automatically discovered concept vocabularies rather than bounded human-annotated attribute sets. They thus seek a different trade-off than our setting, since they favour large-scale concept-set coverage while sacrificing concept detection precision. While restricting them to the 312 ground-truth CUB attributes is necessary to compute human-verified instance-level Precision, doing so forces them into a specialized regime where default optimization budgets are insufficient. Prompted by this concern, we conducted a hyperparameter sweep for both PostHoc and Label-Free CBMs. Please see our resp to Rev. BzHT.
>
> We found that the initially reported extreme value for Label Free was a phenomenon of non-convergence, whereas the results for Post-Hoc CBMs remained largely stagnant. These findings suggest that the underlying class-level optimization objective (which only relies on VLM similarities) fundamentally limits the maximum achievable instance-level precision. Consequently, they yield relatively lower precision and Clarity scores than their instance-level counterparts (especially predictor-based models). We have integrated this context into the corresponding “Other Architectures” discussion in Section 4 and in the appendix to ensure an insightful discussion.
>
> **R3: Human Study Confound (Sparsity vs. Clarity)**
>
> We appreciate this critique regarding a potential visual confound. However, we respectfully wish to clarify that high Clarity models do not exclusively activate few concepts, while low-Clarity models necessarily activate many. For example, the VLM-based $\ell_1$ model achieves an exceptionally high sparsity of 0.974 (activating a mere 2.6% of the bottleneck), yet it exhibits a lower overall Clarity score than the predictor-based $\ell_0$ model, which has a lower sparsity of 0.843.
>
> Nevertheless, following the reviewer’s suggestion, we performed post-hoc Partial Regression and Variance Inflation Factor (VIF) analyses on our human study data. We standardized all variables and modeled aggregate user Trust simultaneously against Clarity and Sparsity to determine whether users were merely reacting to layout density.
>
> When predicting Trust using both variables, we obtain an overall model fit of Adjusted $R^2 = 0.931$. Even after controlling for visual density, Clarity remains a statistically significant predictor of trust ($p = 0.001$), whereas, Sparsity exhibits no statistically detectable unique predictive power in this model ($p = 0.993$).
>
> ### Table 1: Partial Regression
> **Dependent Variable:** Trust | **Adjusted $R^2$:** 0.931 | **F-statistic:** 47.93 ($p = 0.0005$)
>
> | Variable | Standardized Coefficient ($\beta$) | Standard Error | t-statistic | p-value |
> | :--- | :---: | :---: | :---: | :---: |
> | **Clarity** | **0.9741** | **0.131** | **7.453** | **0.001** |
> | Sparsity | 0.0012 | 0.131 | 0.009 | 0.993 |
>
>
> To check the underlying collinearity structure, we calculated the  Variance Inflation Factor (VIF) for our variables:
>
> ### Table 2: Variance Inflation Factor (VIF) Analysis
>
> | Feature | Variance Inflation Factor (VIF) |
> | :--- | :---: |
> | Clarity | **1.7229** |
> | **Sparsity** | **1.7229** |
>
> The low VIF for Sparsity indicates limited collinearity, suggesting that its coefficient estimates are not substantially distorted by multicollinearity. Its near-zero beta coefficient ($\beta = 0.0012$) and large $p$-value ($0.993$), strongly suggest that visual sparsity alone is insufficient to explain the observed trust ratings. Taken together, these findings are consistent with the non-compensatory intuition underlying Clarity: users appear sensitive to semantic quality beyond interface sparsity alone. We have appended this analysis to Section 4 “Human Study” of the revised paper to directly address this potential confound.

---

### Review · Reviewer_KhLw · 2026-05-31

**Summary Of Contributions:**

The paper studies sparsity-aware Concept Bottleneck Models and argues that high predictive accuracy and sparse concept activations do not necessarily imply interpretable or semantically faithful explanations.
It introduces a harmonic-mean metric combining downstream accuracy, sparsity, and concept precision, to penalize models that perform well while relying on semantically incorrect concepts.
The authors evaluate predictor-based and VLM-based CBMs with several sparsity mechanisms, showing that models with similar accuracy can differ substantially.
They further conduct a human study suggesting that the proposed criterion correlates better with perceived trust than accuracy, sparsity, or precision alone.

**Audience:**

Yes

**Audience Explanation:**

This paper explores a relevant research direction that may be inspiring for some people in TMLR's audience.

**Claims And Evidence:**

No

**Claims Explanation:**

> "We introduce Clarity, a novel metric that captures the interplay between downstream performance and the sparsity and precision of concept activations."

The evidence supports Clarity as a reasonable diagnostic summary, but the novelty is limited.
The metric is simply the harmonic mean of three already meaningful quantities: accuracy, sparsity, and precision.
The paper does not sufficiently justify why these criteria need to be collapsed into one scalar rather than reported separately or analyzed through, e.g., Pareto trade-offs.
This paper only focuses on evaluation, not optimization.
In evaluation, sometimes it's more reasonable to report the original scores or even the distributions of scores, not a single scalar value.

---

> Clarity provides "a principled measure of interpretability in concept-based models."

This claim is too strong.
Clarity encodes a particular non-compensatory preference over accuracy, sparsity, and precision, but it is not derived from a principled theory of interpretability.
It is better described as a useful post-hoc diagnostic, not a general measure of interpretability.

---

> "High accuracy is often achieved at the expense of concept precision."

This is broadly supported by the reported experiments.
However, the evidence is limited to a small set of annotated benchmark datasets and selected model families.
The paper should be careful not to overgeneralize beyond these settings.

---

> Clarity distinguishes models that are "right for the right reasons" from those exploiting spurious alignments.

This claim is only partially supported. Clarity can detect whether selected concepts match ground-truth annotations, but this is not the same as proving that the model is causally relying on the right concepts.
A stronger validation would require interventions or counterfactual tests.

**Requested Changes:**

- The paper should more clearly state that its main contribution is an empirical diagnostic for sparse CBMs, not a substantially new CBM architecture or training method. Most of the technical formulation relies on standard CBM/VLM components and standard sparsity mechanisms, while Clarity is a harmonic mean of existing quantities and is used post hoc for evaluation rather than optimized during training.

- The authors should justify why accuracy, sparsity, and concept precision should be collapsed into a single scalar rather than reported separately or analyzed through multi-objective tools. The harmonic mean encodes a particular non-compensatory preference, but the paper sometimes presents it as if it neutrally or uniquely captures interpretability.

- The manuscript should be substantially compressed. Section 3 and many equations are quite verbose. Moving these details to the appendix would make the main contribution easier to identify.

- Improve literature positioning. The paper should better relate CBMs to classical attribute-based zero-shot learning, especially given the image-to-attribute/concept-to-label structure and the use of CUB/SUN attributes. The contribution appears to reinterpret old attribute-alignment issues in the modern CBM/VLM setting, so this lineage should be acknowledged more explicitly.

e.g.,
- Learning To Detect Unseen Object Classes by Between-Class Attribute Transfer
- Zero-Shot Learning - A Comprehensive Evaluation of the Good, the Bad and the Ugly

---

> ### Author Response · Authors · 2026-06-04
>
> We thank the reviewer for their thoughtful and highly constructive critique.
>
> The reviewer raised significant points regarding the positioning of our claims. We have addressed all of these points in our revised manuscript. Below is our point-by-point response outlining the exact updates made.
>
> **R1: Clarifying the Main Contribution as an Empirical Diagnostic Framework and Refining the Tone of Claims**
>
> Indeed, our intention was to introduce an evaluation paradigm rather than a novel training mechanism, and we have tightened the text throughout the entire manuscript to explicitly declare Clarity as a post-hoc diagnostic measure, while toning down the language.
>
> We have updated several key sections to this end.
> Abstract: Adjusted to emphasize: "We introduce Clarity, a post-hoc diagnostic measure that captures the interplay between downstream performance and the sparsity and precision of concept activations."
> Introduction & Related Work: Explicitly states that Clarity acts as a “diagnostic metric for measuring this phenomenon through the joint evaluation of accuracy, sparsity, and concept precision.”
> Section 3.2 (The Clarity Metric): Re-framed to explicitly introduce Clarity as a "new diagnostic metric aiming to quantify the flexibility-interpretability trade-off in concept-based models."
>
>
> **R2: Justifying the Scalar Metric vs. Pareto Trade-Offs**
>
> We thank the reviewer for this comment and we have clarified our motivation in the manuscript. We do not claim that Clarity uniquely or neutrally captures interpretability. Rather, it encodes an explicit design choice: for concept bottleneck models, predictive accuracy, sparsity, and semantic precision are all necessary properties, and severe deficiencies in any one of them undermine the interpretability benefits of the framework. While reporting individual metrics or using Pareto-frontier analysis can provide useful insights, such approaches permit compensatory trade-offs whereby strong performance in one dimension can mask failures in another. Clarity instead adopts a non-compensatory perspective through the harmonic mean, enforcing a strongest penalty on the weakest component. This reflects our view that a CBM with near-perfect accuracy and sparsity but poor semantic alignment should not be considered interpretable. We have revised the text to emphasize that this is a modeling assumption underlying Clarity, rather than a unique definition of interpretability.
>
> We added a relevant discussion in Section 3.2 and Section 4 page 9.
>
>
> **R3: Compressing Section 3**
>
> We have followed the reviewer's advice and streamlined Section 3. We removed standard, verbose background equations describing the standard sparsification mechanism. These details have been compiled into a dedicated section in the Appendix. This architectural compression allows the manuscript to move smoothly from the introduction of the trade-off straight to our proposed framework and primary diagnostic contributions. Section 3 is now a tightly focused description of how the amortized variables are mapped onto our framework and the introduction of the Clarity metric.
>
>
> **R4: Improving Literature Positioning with Classical Zero-Shot Learning (ZSL)**
>
> This is an important historical connection that greatly enhances the foundation of our paper. The concept bottleneck paradigm is heavily linked to classical attribute-based zero-shot learning (ZSL), and the "semantic alignment shift" we document in modern dense VLMs is a direct descendant of the user-defined attribute alignment issues also identified in ZSL frameworks.
>
> To acknowledge this lineage explicitly, we have written a new, dedicated subsection in Section 2 (Related Work) titled "Attribute-based Zero-shot Learning" (Page 3).

---

### Review · Reviewer_BzHT · 2026-06-24

**Summary Of Contributions:**

This paper investigates the trade-off between task performance and interpretability in Concept Bottleneck Models (CBMs). The authors formalize the flexibility-interpretability trade-off, demonstrating how modern CBM architectures can achieve high predictive performance by relying on semantically incorrect concept representations. To quantify this, the paper introduces "Clarity," a novel post-hoc diagnostic metric defined as the harmonic mean of dataset-level sparsity, concept precision, and downstream classification accuracy. The authors also introduce amortized sparsity-aware CBM formulations using different sparsity mechanisms and evaluate them against various baselines.

**Additional Comments:**

N/A

**Audience:**

Yes

**Audience Explanation:**

The proposed Clarity metric and the systematic analysis of the flexibility-interpretability trade-off offer valuable insights for researchers working on representation learning, vision-language models, and trustworthy machine learning.

**Claims And Evidence:**

Yes

**Claims Explanation:**

The claims are well supported by the experimental evaluations, theoretical formulations, and the human study.

**Requested Changes:**

1. Explicitly address the edge case in Equation 11 where the denominator becomes zero if no concepts are active for a given instance. Clarify how per-instance Precision is assigned in this scenario (e.g., whether it defaults to zero, or if the instance is excluded) and discuss how this choice affects the dataset-level Precision and the final Clarity metric.
2. Figure 2 provides insightful Clarity-Accuracy curves. It would strengthen the empirical results to briefly discuss the variance or stability of these curves across different random seeds, ensuring that the threshold selection strategy proposed in Equation 15 is robust.
3. ** In Table 4, you compare the proposed methods against global sparsity baselines (e.g., Post-Hoc CBM, Label-Free CBM). Since these baselines were originally designed for large, automatically discovered vocabularies, replacing their concept sets with ground-truth CUB attributes might inherently handicap their precision. Please add a brief discussion clarifying whether their hyperparameters were tuned for this bounded setting, and how this mismatch might affect the fairness of the comparison.

---

> ### Author Response · Authors · 2026-06-25
>
> We thank the reviewer for their insightful comments and for highlighting that our approach offers valuable insights for researchers in the related fields. Below, we address the concerns raised:
>
> 1. On the Edge Case in Equation 11 (Zero Denominator)
>
> We thank the reviewer for pointing out this critical boundary case. We have updated Section 3.2 immediately following Equation 11 to formally clarify our handling of this scenario. Specifically, if a model predicts no active concepts for an instance, we explicitly assign an instance-level precision of $0$. Because our dataset-level Precision is a macro-average over all instances, these zero-valued assignments directly lower the aggregate Precision score and subsequently the resulting Clarity. We intentionally chose not to exclude these zero-concept instances from the evaluation since it can potentially lead to unrepresentative and artificially inflated precision scores for over-sparsified or conservative models. For example, a model that remains entirely "silent" across a large portion of the dataset could present a misleadingly high precision if evaluated only on the few instances where it happened to activate concepts. Retaining these instances and assigning them a score of $0$ ensures that our evaluation framework provides an accurate, comprehensive diagnostic summary of the model’s true coverage across the entire dataset.
>
> 2. On the Stability of Figure 2 Curves and Equation 15 Thresholding
>
> We completely agree that establishing the statistical stability of our thresholding strategy is vital for its practical utility.  To this end, we performed five independent runs for VLM-based Bernoulli models.
>
> Our empirical tracking shows that the standard deviation across different runs is exceptionally low. For the ViT-B/16 VLM-based Bernoulli models, we observe $\sigma_{\text{acc}} = 0.60$, $\sigma_{\text{prec}} = 0.0078$, $\sigma_{\text{sparsity}} = 0.0010$, and $\sigma_{\text{clarity}} = 0.0107$. For the larger ViT-L/14 variants, these variances remain similarly bounded at $\sigma_{\text{acc}} = 0.23$, $\sigma_{\text{prec}} = 0.0053$, $\sigma_{\text{sparsity}} = 0.0005$, and $\sigma_{\text{clarity}} = 0.0077$. This high reproducibility suggests that stochastic optimization noise does not significantly impact the macro-behavior of the models. Consequently, the threshold selection strategy proposed in Equation 15 converges reliably to the exact same operational regions across runs, confirming its robustness.We added these variability results in the appendix.
>
> 3. On Baseline Hyperparameter Tuning and Fairness (Table 4)
>
> Following the reviewers' concerns, we performed additional experiments to assess how the hyperparameters of the baseline models affect their resulting Clarity. Specifically, for Post-Hoc CBMs, the two main hyperparameters to tune are the regularization strength of the Elastic Net ($\lambda$) and the balancing term between the $L_1$ and $L_2$ penalties ($\alpha$). We considered three different levels of regularization strength, $\lambda \in [10^{-5}, 10^{-4}, 10^{-3}]$, and two configurations for the balancing term: the default of the original implementation ($\alpha = 0.99$) and the default for the standard scikit-learn package ($\alpha = 0.15$). Even though we observed significantly different behaviors in terms of bottleneck sparsity and downstream classification accuracy, the concept precision remained largely stagnant around 10–11%.
>
> For Label-Free CBMs, we evaluated different optimization settings, sweeping across combinations of projection layer steps ($1000$ and $5000$) and SAGA solver iterations ($1000$ and $5000$). We observed that the default hyperparameters ($1000$ steps) did not allow for proper convergence when forcing the model to map onto a strictly bounded, ground-truth vocabulary space. By increasing these optimization budgets to $5000$ projection steps and $5000$ solver iterations, we facilitated full convergence, which managed to yield significantly better results, noticeably increasing its concept precision from its initial baseline up to $0.1690$. Despite this clear convergence improvement, we still observe that the underlying class-level optimization objective based on the VLM capabilities for discovering concepts, fundamentally limits the maximum achievable instance-level precision, yielding relatively low precision scores.
>
> We have added a discussion in the appendix and have incorporated these newly optimized performance values into Table 4 and all corresponding tracking metrics throughout the manuscript.

---

### Author Response · Authors · 2026-07-13

Dear Action Editor and Reviewers,

As we approach the end of the discussion period, we wanted to kindly follow up regarding our previous responses and the updated manuscript.

We hope our revisions and clarifications have successfully addressed your feedback. If there are any remaining concerns, or if further clarifications would be helpful, please let us know. We remain very open to further discussion and would be happy to provide any additional information needed.

Thank you again for your time, careful consideration, and constructive feedback.

Best regards,
the Authors